# On the Robustness of Deep Clustering Models: Adversarial Attacks and Defenses

**Anshuman Chhabra**[*], **Ashwin Sekhari**[*], **and Prasant Mohapatra**
Department of Computer Science
University of California
Davis, CA 95616
{chhabra,asekhari,pmohapatra}@ucdavis.edu

## Abstract

Clustering models constitute a class of unsupervised machine learning methods which are used in a number of application pipelines, and play a vital role in modern data science. With recent advancements in deep learning– deep clustering models have emerged as the current state-of-the-art over traditional clustering approaches, especially for high-dimensional image datasets. While traditional clustering approaches have been analyzed from a robustness perspective, no prior work has investigated adversarial attacks and robustness for deep clustering models in a principled manner. To bridge this gap, we propose a blackbox attack using Generative Adversarial Networks (GANs) where the adversary does not know which deep clustering model is being used, but can query it for outputs. We analyze our attack against multiple state-of-the-art deep clustering models and real-world datasets, and find that it is highly successful. We then employ some natural unsupervised defense approaches, but find that these are unable to mitigate our attack. Finally, we attack `Face++`, a production-level face clustering API service, and find that we can significantly reduce its performance as well. Through this work, we thus aim to motivate the need for *truly* robust deep clustering models.

## 1 Introduction

Clustering models are utilized in many data-driven applications to group similar samples together, and dissimilar samples separately. They constitute a powerful class of unsupervised Machine Learning (ML) models which can be employed in many cases where labels for data samples are either hard (or impossible) to obtain. Note that there are a multitude of different approaches to accomplishing the aforementioned clustering task, and an important differentiation can be made between *traditional* and *deep* clustering approaches.

Traditional clustering generally aims to minimize a clustering objective function defined using a given distance metric [1]. These include approaches such as k-means [2], DBSCAN [3], spectral methods [4], among others. Such models generally fail to perform satisfactorily on high-dimensional data (i.e., image datasets), or incur huge computational costs that make the problem intractable [5]. To improve upon these approaches, initial deep clustering models sought to decompose the high-dimensional data to a *cluster-friendly* low-dimensional representation using deep neural networks. Clustering was then undertaken on this latent space representation [6, 7]. Since then, deep clustering models have become considerably advanced, with state-of-the-art models outperforming traditional clustering models by significant margins on a number of real-world datasets [8, 9].

Despite these successes, deep clustering models have not been sufficiently analyzed from a *robustness* perspective. Recently, traditional clustering models have been shown to be vulnerable to adversarial

---

[*]Equal contribution.

36th Conference on Neural Information Processing Systems (NeurIPS 2022).

attacks that can reduce clustering performance significantly [10, 11][2]. However, no such adversarial attacks exist for deep clustering methods. Furthermore, no work investigates *generalized blackbox* attacks, where the adversary has zero knowledge of the deep clustering model being used. This is the most realistic setting under which a malicious adversary could aim to disrupt the working of these models. While there is a multitude of work in this domain for supervised learning models, deep clustering approaches have not received the same attention from the community.

The closest work to ours proposes robust deep clustering [9, 12], by retraining models with adversarially perturbed inputs to improve clustering performance. However, this line of work has many shortcomings: 1) it lacks fundamental analysis on attacks specific to deep clustering models (for e.g., the state-of-the-art robust deep clustering model RUC [9] only considers images perturbed via FGSM/BIM [13, 14] attacks, which are common attack approaches for *supervised* learning), 2) no clearly defined threat models for the adversary are proposed[3], and 3) there is no transferability [15] analysis. Thus in this work we seek to bridge this gap by proposing generalized *blackbox* attacks that operate at the input space under a well-defined adversarial threat model. We also conduct empirical analysis to observe how effectively adversarial samples transfer between different clustering models.

We utilize Generative Adversarial Networks (GANs) [16] for our attack, inspired by previous approaches (AdvGAN [17], AdvGAN++ [18], etc) for supervised learning. We also utilize a number of defense approaches (essentially deep learning based anomaly detection [19] and state-of-the-art "robust" deep clustering models [9]) to determine if our adversarial samples can be *mitigated* by an informed de-

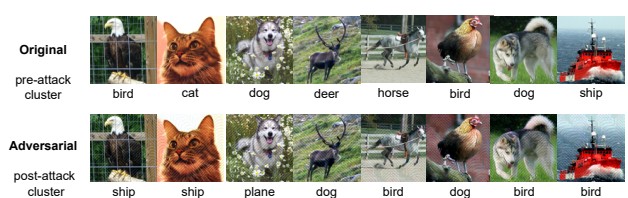

Figure 1: Adversarial samples generated by our attack (first 4 image pairs from the left correspond to SPICE and the others to RUC).

fender. One of the major findings of our work is that these approaches are unable to prevent our adversarial attacks. Through our work, we seek to promote the development of better defenses for adversarial attacks against deep clustering.

Finally, to truly showcase how powerful our attacks can be, we attack a production-level ML-as-a-Service (MLaaS) API that performs a clustering task (and possibly utilizes deep clustering models in the backend). We find that our attack can also significantly reduce the functioning of such MLaaS clustering API services. To summarize, we make the following contributions:

• We propose the first blackbox adversarial attack against deep clustering models. We show that our attacks can significantly reduce the performance of these models while requiring a minimal number of queries. We also undertake a transferability analysis to demonstrate the magnitude of our attack.

• We undertake a thorough experimental analysis of most state-of-the-art (SOTA) deep clustering models, on a number of real-world datasets, such as CIFAR-10 [20], CIFAR-100 [21], and STL-10 [22] which shows that our attack is applicable across a number of models and datasets.

• We show that existing (unsupervised) defense approaches (such as anomaly detection and robustness via adversarial retraining) cannot thwart our attack samples, thus prompting the need for better defense approaches for adversarial attacks against deep clustering models.

• We also attack a production-level MLaaS clustering API to showcase the extent of our attack. We find that our attack is highly successful for this real-world task, underscoring the need for making deep clustering models truly robust.

Figure 1 shows some of the adversarial samples generated by our attack for the SPICE [8] and RUC [9] deep clustering models on the STL-10 dataset, and their corresponding pre-attack and post-attack predicted cluster/class labels[4]. To the human eye, these samples appear indistinguishable from each

---

[2]These attacks cannot be used for deep clustering because: 1) they employ computationally intensive optimizers that use exhaustive search and hence, make the attack too expensive for high-dimensional data, 2) they are designed for traditional clustering and are *training-time* attacks, but, deep clustering models are deployed *frozen* for inference, requiring a *test-time* attack.

[3]For example, for RUC [9], it is implicitly assumed that the adversary has knowledge of the dataset as well as ground truth labels, and will attack using supervised whitebox attacks.

[4]*Class* labels are inferred by taking the majority from the ground truth labels for samples in that cluster.

other, but the model clusters them incorrectly after the attack. As we will show in later sections, our attack on SPICE for STL-10 results in a 83.9% (67.9% for RUC) decrease in clustering utility measured according to the NMI [23] metric[5].

## 2    Related Works

**Deep Clustering.** To leverage clustering algorithms on high-dimensional data, early work on deep clustering [6, 7], aimed to learn a latent low-dimensional *cluster-friendly* representation that could then be clustered on. This task was achieved using AutoEncoders (AEs) [7, 24, 25], Variational AEs [26, 27], or GANs [28–30]. More recently, current SOTA deep clustering models employ self-supervised and contrastive learning instead of the earlier AE/GAN based approaches to perform clustering [8, 9, 31]. We consider these in the paper due to their superlative performance.

**Training-time Adversarial Attacks Against Clustering.** Recently, a few works have proposed blackbox adversarial attacks against clustering [10, 11]. These seek to poison a small number of samples in the input data, so that when clustering is undertaken on the poisoned dataset, other unperturbed samples change cluster memberships. This problem is significantly different from ours– the traditional clustering algorithm *retrains* on the poisoned input data constituting an attack at training time. For deep clustering, the model does not train again once deployed, so we have to generate adversarial images that the model misclusters at *test time*. There have also been whitebox attacks proposed for traditional clustering, but these can only be employed for the specific traditional clustering algorithm considered [32–34].

**Attacks/Defenses in Supervised Learning.** The closest attacks to ours in supervised learning are *score-based* blackbox attacks where the adversary has access to the softmax probabilities outputted by the model [35]. A number of such attacks have been proposed: NES [36], SPSA [37], among others. These cannot be applied in their original formulation as the attack optimization expects ground truth reference labels. Even with simple modifications to the loss (such as using predicted labels as ground truth) these attacks do not work as successfully as our proposed GAN attack (refer to Appendix F.1 for a detailed discussion and Appendix E for empirical justification). Similar issues make supervised defenses inapplicable for our setting– defense strategies need to be truly unsupervised to work in the clustering context, and outputs of deep clustering models might not possess a one-to-one mapping with ground truth labels, causing further problems. We provide a more detailed discussion on the inapplicability of these defense methods in Appendix F.2.

**Robust Deep Clustering.** A natural in-processing defense approach to preventing adversarial attacks on deep neural networks is to utilize some form of *adversarial retraining* which seeks to jointly optimize an adversarial objective along with the original loss function used to train the network. This technique has been shown to vastly improve the adversarial robustness of traditional deep neural networks [38, 39]. In the context of deep clustering, to the best of our knowledge, there are two works that aim to make models robust to adversarial noise in this manner[6]: RUC [9] and ALRDC [12]. Moreover, RUC is considered to be the SOTA *robust* deep clustering model. Even in terms of performance metrics (NMI, ACC, ARI) it is second only to the SOTA model SPICE [43].

## 3    Preliminaries and Notation

In this section, we introduce notation and preliminary knowledge regarding the clustering task and the proposed adversarial attack. Note that for a matrix $A$ of size $u \times v$, we can index into row $i$ as $A_i, \forall i \in [u]$, and index into a single value at row $i$ and column $j$ as $A_{i,j}, \forall i \in [u], j \in [v]$. Moreover, $||.||$ denotes the Euclidean (2-norm) norm of a vector.

**Deep Clustering.** Since we are proposing blackbox attacks in our paper, we will be defining deep clustering models in a more generalized manner that abstracts their inner functioning. We defer the reader to [5] for more details on the models analyzed in the paper. A deep clustering model is denoted as $\mathcal{C}$ and operates on samples of the given dataset $X$ consisting of $n$ samples and maps them

---

[5]Similar results for SPICE hold for CIFAR-100 with 78.9% (58.2% for RUC) decrease in NMI, and for CIFAR-10 with 72.9% (68.7% for RUC) decrease in NMI.

[6][40–42] refer to "robustness" in their work but their definition is not the traditional notion of adversarial robustness and they do not incorporate adversarial retraining.

to one of $k$ clusters. As deep clustering models utilize deep neural networks internally, they generate a set of softmax probabilities indicating cluster memberships, denoted as $M \in [0,1]^{n \times k}$. From this, the cluster labels can be obtained by taking the maximum of the cluster probabilities. For a dataset sample $X_i \in X$, the cluster label can be obtained as $l = \text{argmax}_{j \in [k]}\{M_{i,j}\}$. We denote the vector of cluster labels computed in this manner as $L \in \mathbb{N}^{n \times 1}$.

Note that unlike supervised learning problems, the ground truth labels $Y \in \mathbb{N}^{n \times 1}$ are not utilized for training the model in deep clustering. However, these ground truth labels are used for evaluating the performance of the model. That is, the output cluster labels $L$ are evaluated in comparison with the ground truth $Y$. Performance metrics such as Normalized Mutual Information (NMI) [23], Adjusted Rand Index (ARI) [44], and Unsupervised Accuracy (ACC) [45] are commonly used in deep clustering literature for this purpose. These metrics are defined analytically in Appendix A.6.

**Threat Model.** We now define the threat model for the adversary:

1. The attacker has knowledge of the dataset $X$. This is a commonly made assumption in adversarial attacks against traditional clustering literature [10, 11].

2. The attacker carries out a blackbox attack and does not know which deep clustering model $\mathcal{C}$ is being used, but can query it to observe $M$ (softmax cluster memberships), and hence, $L$[7]. In adversarial attacks on (un)supervised models, this is often what constitutes a blackbox attack [46–49].

3. The goal of the attack is to provide minimally perturbed images (there is a noise threshold that cannot be exceeded) as input to the deep clustering model. Upon doing so, the model should *miscluster* these samples and the performance measured via the evaluation metrics (ACC, NMI, ARI, etc) should significantly reduce post the attack.

**Adversary's Objective.** The goal of the attacker is to input adversarial images to the model and lead to a performance decrease, measured using metrics such as NMI, ACC, and ARI. We cannot directly optimize post-attack $M$ or $L$ as the adversary does not possess ground truth labels. Moreover, cluster labels generated by the model may not be the same as the ground truth labels, thus requiring some notion of a mapping function, further complicating the problem. Instead we will indirectly achieve this goal through another objective function.

Let an input image be $X_i \in X$ and through a single query the cluster probabilities $M_i$ can be obtained before the attack. To attack, the adversary will introduce a carefully crafted perturbation/noise $\delta$ specific to this sample. The attacker queries the deep clustering model and obtains the set of cluster probabilities for this sample. Abusing notation slightly, these are denoted as $\mathcal{C}(X_i + \delta)$.

Assuming the original unperturbed cluster labels/probabilities accurately depict the cluster representing the sample's ground truth label, the attacker can simply generate the adversarial noise via the following optimization problem:

$$\max_{\delta} \quad ||M_i - \mathcal{C}(X_i + \delta)|| \quad \text{s.t.} \quad ||\delta|| \leq \epsilon \qquad \text{(Attacker's Optimization)}$$

Here the constraint with $\epsilon$ is simply to ensure that the adversarial sample does not have unbounded noise and remains *realistic* to a human observer. The objective function above ensures that the cluster probabilities post the attack $\mathcal{C}(X_i + \delta)$ are as distinct as possible to the cluster probabilities $M_i$ obtained before the attack. Then, assuming that prior to the attack the cluster labels were the correct representative of the ground truth label $Y_i$[8], the deep clustering model will miscluster the sample after the attack and performance (NMI, ACC, ARI) should reduce. If the adversary introduces many such adversarial samples, the performance can thus be significantly affected[9].

---

[7]If only $L$ can be observed it is a *decision-based* attack; if $M$ can be observed it is a *score-based* attack [35].

[8]This is true for most samples in the dataset as clustering performance of the model prior to the attack is assumed to be superlative, otherwise there is no reason to attack.

[9]Note that this is an *untargeted* attack. One could also consider *targeted* attacks by replacing $M_i$ in the objective with a $k$ length vector where the target cluster entry is 1, and all other entries are 0.

# 4 Attacking Deep Clustering Models

## 4.1 Blackbox Attack Using GANs

We utilize a simple GAN based architecture for generating the adversarial perturbation $\delta$ and solving the attack problem delineated in the previous section. There are many different variations to using GANs for generating adversarial samples, such as AdvGAN [17], AdvGAN++ [18], WPAdvGAN [50], CycleAdvGAN [26], among many others. For our attack, we employ a vanilla GAN architecture consisting of deep neural networks for both the Generator and Discriminator, similar to AdvGAN.

We utilize the Generator model $\mathcal{G}$ to generate the adversarial perturbation $\delta$ for a given input image $X_i \in X$, i.e., $\mathcal{G}(X_i) \to \delta$. The Discriminator model $\mathcal{D}$ plays a similar role as for the original GAN model [16] as it aims to ensure that the perturbed image is similar to the distribution of input images.

We then rewrite our attack optimization problem in the context of the GAN architecture. The loss for the attack objective can be written directly as in the optimization problem:

$$\mathcal{L}_{\text{attack}} := \mathbb{E}_{X_i} ||M_i - \mathcal{C}(X_i + \mathcal{G}(X_i))||$$

And we can simply reformulate the constraint on the norm of the adversarial noise as:

$$\mathcal{L}_{\text{constraint}} := \mathbb{E}_{X_i} \min\{\epsilon - ||\mathcal{G}(X_i)||, 0\}$$

We also write the vanilla minimax GAN loss [16] as:

$$\mathcal{L} := \mathbb{E}_{X_i}[\log(\mathcal{D}(X_i)) + \log(1 - \mathcal{D}(X_i + \mathcal{G}(X_i)))]$$

To train the Generator $\mathcal{G}$ and Discriminator $\mathcal{D}$, we then optimize these combined losses by solving the following saddle-point problem (where $\alpha_a, \alpha_c$ are hyperparameters to control tradeoff):

$$\max_{\mathcal{D}} \min_{\mathcal{G}} \{\mathcal{L} - \alpha_a \mathcal{L}_{\text{attack}} - \alpha_c \mathcal{L}_{\text{constraint}}\}$$

Upon obtaining the trained Generator $\mathcal{G}$, we can generate the adversarial perturbation as $\delta = \mathcal{G}(X_i)$ for any image $X_i \in X$ provided as input. We then provide these adversarial images $X_i + \delta$ as input to the pre-trained deep clustering model $\mathcal{C}$ to obtain cluster membership confidence scores as $M_i' = \mathcal{C}(X_i + \delta)$ after the attack. From $M_i$ we know the original cluster label as $L_i = \text{argmax}_{j \in [k]} M_{i,j}$ and similarly, we can obtain the cluster label of the adversarial image as $L_i' = \text{argmax}_{j \in [k]} M_{i,j}'$.

If the optimization problem was solved successfully and the distance between $M_i$ and $\mathcal{C}(X_i + \delta)$ is sufficiently large enough, we can have $L_i \neq L_i'$. Moreover, assuming that a mapping function $\phi$ exists that maps the output cluster labels to the ground truth labeling, we know that: $\phi(L_i) = Y_i$. Since $L_i \neq L_i'$, we can conclude that $\phi(L_i') \neq Y_i$ leading to misclustering and a drop in performance.

**Remark.** Note that since our attack objective is indirectly formulated, it is possible that $M_i$ and $M_i'$ are not sufficiently far apart even after the attack, leading to $L_i = L_i'$. However, as we find and our experiments show, this does not happen frequently in practice as our blackbox attack is highly successful at generating adversarial samples that disrupt the performance of all deep clustering models considered in the paper (Section 6.1). We have considered the following open-source deep clustering models in experiments: CC [31], SPICE [8], SCAN [51], MiCE [52], NNM [53], RUC [9].

## 4.2 Attacking `Face++`: A Production-level MLaaS Clustering API

To showcase the disruptive capability of our attacks, we attack `Face++`[10], which provides an extremely well-performing face clustering REST API service. At a high-level, the API performs a basic face clustering task where it takes in as input a face dataset and seeks to cluster images belonging to the same person together. We do not know the clustering algorithm/model being utilized in the backend. Note that there is one significant difference here compared to the threat model for open-source deep clustering models previously considered– we do not have access to cluster memberships, but just the final set of labels. To overcome this problem, we attack `Face++` by training a *surrogate* open-source deep clustering model on the dataset and then generate adversarial samples for this model using our

---

[10]https://www.faceplusplus.com/photo-album-clustering/

GAN based attack pipeline. We then use these adversarial samples as input to the `Face++` API to conduct the attack. Hence, our attack constitutes a *transferability* attack via a *surrogate* model.

The `Face++` API service functions as follows: we first create a new *face album* using the `createAlbum` endpoint which gives us a `facealbum_token` as a response. Using this token, we add the images for our dataset using the `addimage` endpoint, and then call the `groupFace` endpoint to perform the clustering task. Finally, we obtain cluster labels for each image using the `getAlbumDetail` endpoint, which returns `group_ids` as the integer cluster labels. We provide the results of our attack and experiment details in Section 6.3 as well as additional implementation details in Appendix C.

## 5 Mitigating the Attack: Possible Defense Approaches

**In-processing Defense: Robust Deep Clustering.** Adversarially retrained models (where the learning process incorporates a joint adversarial loss along with the original loss) have been shown to considerably mitigate adversarial attacks, and thus constitute a natural in-processing defense approach. In the context of deep clustering, RUC and ALRDC utilize adversarial learning to improve model robustness. However, we only consider RUC in this paper for the following reasons: 1) RUC is an add-on module, and can be applied to any deep clustering model, 2) in contrast, the ALRDC approach specifically works only to make the latent space robust to perturbations (which most deep clustering models considered in our paper do not possess), restricting its use, and 3) unlike RUC, from our early experiments with ALRDC we found that it did not work well with high-dimensional real-world image datasets such as CIFAR-10/CIFAR-100/STL-10, which we have considered in this paper[11].

In our experiments in Section 6.2, we show that using our GAN based attack (Section 4.1) and RUC as the deep clustering model $\mathcal{C}$, we can disrupt its performance significantly as well. Once the generator has learnt how to generate adversarial noise specific to RUC, it can easily degrade RUC's clustering ability. These results clearly indicate that there is a deficiency in current robust deep clustering strategies, and the problem requires different solutions[12].

**Pre-processing Defense: Deep Learning Based Anomaly Detection.** With advancements in deep learning, the long-standing field of anomaly/outlier detection has also seen major advancements. Further, due to the unsupervised nature of the deep clustering task (no labels), anomaly detection is a suitable pre-processing approach for detecting adversarial samples. In particular, most deep learning based anomaly detection models are trained on specific datasets and can detect out-of-distribution samples with extremely high precision. For our experiments, we use a recently proposed self-supervised deep anomaly detection approach SSD [19] which achieves state-of-the-art performance on benchmarks when labeled training data is not present [56].

While a detailed description of SSD is beyond the scope of our work, the authors employ contrastive self-supervised representation learning combined with a Mahalanobis distance based threshold detector in the feature space to detect anomalies. In our experiments in Section 6.2 we show that even SSD is unable to detect a large majority of our adversarial samples. We further show that this is likely due to the distribution of our adversarial samples in space, as they mimic the original samples fairly well due to the norm constraint on the generator's output. We use Principal Component Analysis (PCA) [57] to analyze the adversarial and benign samples and find that their principal components tend to be very similar. In this regard, anomaly detection approaches are also unable to detect most of our generated adversarial samples (less than 1% in the best case, and $\approx 14\%$ on average!).

## 6 Experiments and Results

### 6.1 Blackbox Attack Against Deep Clustering Models

For our experiments on the open-source models, we utilize the following real-world image datasets: CIFAR-10 [20], CIFAR-100 [21], and STL-10 [22]. These are commonly utilized in a majority of deep clustering literature. For the models, we consider a number of state-of-the-art deep clustering models: SPICE [8], SCAN [51], NNM [53], MiCE [52], and CC [31]. We also consider RUC [9], but discuss

---

[11]In the ALRDC paper only MNIST [54] and Fashion-MNIST [55] were considered in their experiments.

[12]One simple but compute-intensive solution could be to jointly minimize the deep clustering loss on adversarial inputs by using our trained adversarial noise generators, similar to traditional adversarial retraining.

its results as part of the defense Section 6.2. Notably, SPICE and RUC (add-on to SCAN) are SOTA #1 and #2 for performance on the aforementioned datasets. We show a large number of adversarial images generated by our attack model and provide additional experiment details in Appendix A.

Table 1: Pre-attack and post-attack performance for open-source deep clustering models.

| Model | | CIFAR-10 | | | CIFAR-100 | | | STL-10 | | |
|---|---|---|---|---|---|---|---|---|---|---|
| | | NMI | ARI | ACC | NMI | ARI | ACC | NMI | ARI | ACC |
| CC | Pre-attack | 0.70 | 0.64 | 0.79 | 0.43 | 0.27 | 0.43 | 0.76 | 0.73 | 0.85 |
| | Post-attack | **0.01** | **0.00** | **0.10** | **0.03** | **0.00** | **0.07** | **0.03** | **0.00** | **0.12** |
| MiCE | Pre-attack | 0.73 | 0.69 | 0.83 | 0.45 | 0.29 | 0.44 | 0.56 | 0.46 | 0.62 |
| | Post-attack | **0.20** | **0.12** | **0.44** | **0.15** | **0.04** | **0.20** | **0.30** | **0.20** | **0.43** |
| NNM | Pre-attack | 0.75 | 0.71 | 0.84 | 0.48 | 0.32 | 0.48 | 0.70 | 0.65 | 0.81 |
| | Post-attack | **0.30** | **0.09** | **0.41** | **0.18** | **0.01** | **0.15** | **0.22** | **0.07** | **0.25** |
| SCAN | Pre-attack | 0.71 | 0.66 | 0.82 | 0.49 | 0.33 | 0.51 | 0.67 | 0.62 | 0.79 |
| | Post-attack | **0.27** | **0.06** | **0.48** | **0.10** | **0.01** | **0.14** | **0.16** | **0.02** | **0.22** |
| SPICE | Pre-attack | 0.85 | 0.84 | 0.92 | 0.57 | 0.39 | 0.54 | 0.87 | 0.87 | 0.94 |
| | Post-attack | **0.23** | **0.10** | **0.36** | **0.12** | **0.02** | **0.15** | **0.14** | **0.00** | **0.16** |
| RUC | Pre-attack | 0.83 | 0.81 | 0.90 | 0.55 | 0.39 | 0.53 | 0.78 | 0.74 | 0.87 |
| | Post-attack | **0.26** | **0.08** | **0.33** | **0.23** | **0.05** | **0.26** | **0.25** | **0.07** | **0.30** |

As part of our experiment, we generate adversarial images (adversarial set) using our attack for all the images in the original test set. We show the performance metrics (ACC, NMI, ARI), pre-attack (original images) and post-attack (adversarial images) for all the aforementioned models and datasets in Table 1. Quite evidently, it can be observed that performance for all models is significantly reduced post the attack. Note that since the GAN network generates fixed noise for the same input, there is no variance in the obtained results. We have presented hyperparameter ($\alpha_a, \alpha_c, \epsilon$, etc.) values for all experiments in this section in Appendix A.1 due to space limitations.

**Effects of the Attack.** For all the deep clustering models, we find a general trend emerges in how the models' post-attack clusters differ from the clusters generated on the original test set. In particular, performance on the benign images is generally very good– this can be observed in the confusion matrix for the SPICE model and STL-10 dataset in Fig 2a. Note that most of the clusters are correctly defined and there are a few miclusterings. However, for the adversarial images, we notice that there is a complete *clustering breakdown*– most images tend to get lumped in a small number of select clusters, and the remaining few images create small-sized clusters. This can also be seen in the confusion matrix in Fig 2b for SPICE on adversarial STL-10 images. Most of the adversarial images are clustered as part of the "cat" cluster, despite being clustered accurately before the attack. This trend is prevalent for the attack on all the models, and tends to be more drastic for less performant deep clustering models, such as CC. This can be seen in the confusion matrices shown in Figs 2c and 2d. We present the confusion matrices for the other models in Appendix A.3 due to limited space.

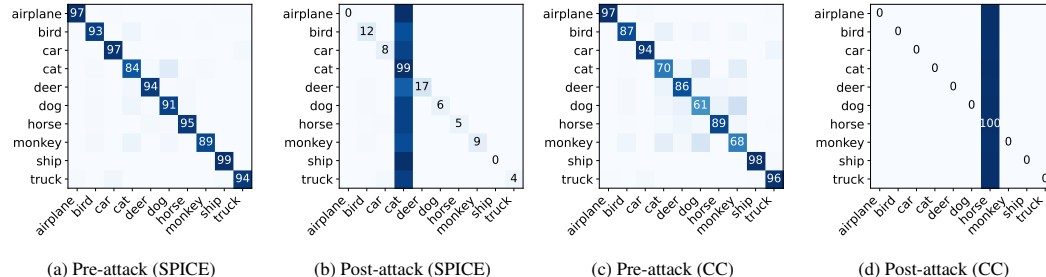

(a) Pre-attack (SPICE)     (b) Post-attack (SPICE)     (c) Pre-attack (CC)     (d) Post-attack (CC)

Figure 2: Confusion matrices showcasing the effect of the attack for the SPICE/CC models on STL-10 dataset.

A major takeaway from these results is that while deep clustering models tend to work very well on "clean" data, simple adversarial samples exist that can completely degrade performance. It is thus imperative that deep clustering model designers evaluate their models against adversarial samples.

**Query Complexity and Perturbation Analysis.** We also measure *query complexity* of our approach to analyze the cost associated with carrying out our attack in the real world. In this blackbox attack scenario, the query complexity is defined as the number of times the deep clustering model is queried

Table 2: Query complexity of the attack.

| Model | CIFAR-10 | CIFAR-100 | STL-10 |
|---|---|---|---|
| CC | 5148 | 5382 | 2150 |
| MiCE | 2820 | 3142 | 2244 |
| NNM | 2320 | 11360 | 4660 |
| SCAN | 3100 | 2200 | 3080 |
| RUC | 3500 | 4000 | 3380 |
| SPICE | 2320 | 7520 | 2304 |

by the adversary. Thus, we can measure the number of times the Generator $\mathcal{G}$ queries the clustering model $\mathcal{C}$ with an input batch, before the loss of the GAN network converges. We present these results in Table 2 where the batch size is 256. It can be seen that the query complexity of our attack is quite minimal for all models and datasets. In particular, the values obtained for our method are comparable to the query complexity rates obtained by existing blackbox attacks against supervised learning models [46, 48]. Moreover, a smaller query complexity is doubly beneficial in our case, because once the generator has been trained, we can use it to generate the optimal perturbation for any images that belong to that input distribution without requiring any additional queries to the clustering model.

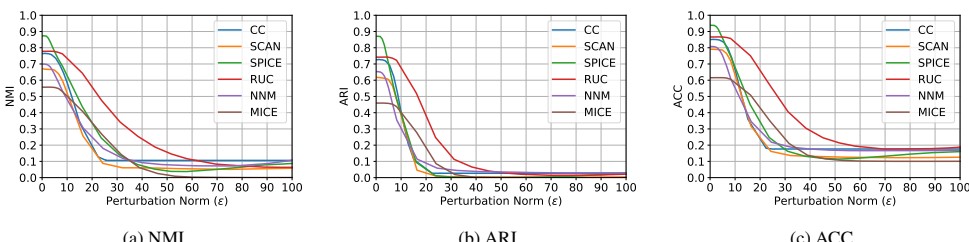

| (a) NMI | (b) ARI | (c) ACC |

Figure 3: Performance as a function of the adversarial perturbation norm for STL-10.

We also analyze the effect of varying the noise penalty (via $\epsilon$) on the extent to which the attack degrades the performance of the deep clustering model. We depict the NMI, ARI, ACC for all the models on the STL-10 dataset as a function of the norm of the generated perturbation in Fig 3. It can be observed that as the norm threshold is increased the attack becomes more successful and the performance of the models is worsened, eventually plateauing close to 0. To conserve space, we defer these results with similar trends for the other datasets to Appendix A.4.

**Transferability Analysis.** We undertake a transferability analysis to see whether adversarial samples generated for one model transfer to other deep clustering models. We present these results in Fig 4 for all our datasets as transferability matrices. In these, we show the post-attack NMI for source and target models, along with the pre-attack NMI for each of the deep clustering models. It can be seen that the overall transferability of most adversarial samples is high. Note that SPICE (SOTA #1) is in general a much better deep clustering model than CC in terms of performance. Interestingly however, we can see that SPICE's adversarial samples do not transfer well to CC and vice versa. It would be interesting to investigate the reasons for this occurrence in future theoretical or empirical work. Due to space limitations, we show the transferability matrices with ACC and ARI in Appendix A.5.

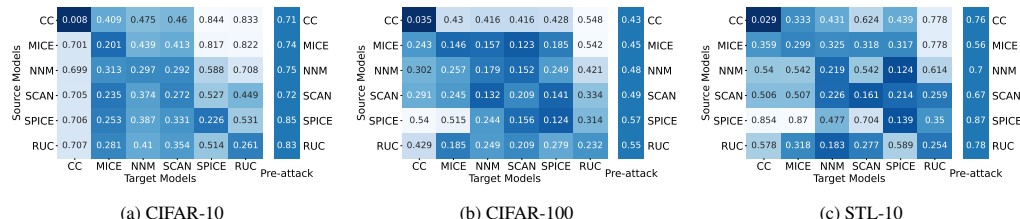

| (a) CIFAR-10 | (b) CIFAR-100 | (c) STL-10 |

Figure 4: Transferability results showcasing post-attack (and pre-attack) NMI for different source/target models.

## 6.2 Defense Approaches

**Robust Deep Clustering.** As mentioned before, we utilize the RUC add-on module as a first defense against our attack. As in the original paper, we utilize RUC with SCAN as the deep clustering model and then carry out our GAN based attack against it. Similar to the experiments on the open-source models in Section 6.1, we measure the performance on a benign test set and then on a corresponding adversarial set generated using our GAN attack. We find that our attack is also successful against RUC, thus proving that RUC is not entirely robust to adversarial noise. This is shown in Table 1. It can be seen that RUC is not a satisfactory defense approach for our attack. However, note that the robustness of RUC is somewhat observable in our perturbation norm versus performance analysis shown in Fig 3. In general, it takes higher norm thresholds to reduce the performance of RUC, although it is achievable without adding too much noise to the images. For the transferability analysis shown in Fig 4, we find RUC behaves similar to SPICE, as RUC's adversarial samples are highly transferable across all models and datasets except for CC on the CIFAR-10 and CIFAR-100 datasets.

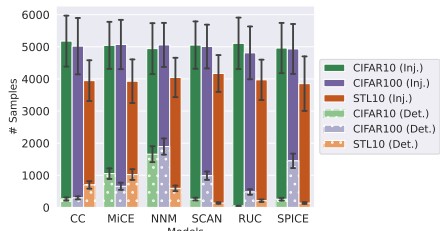

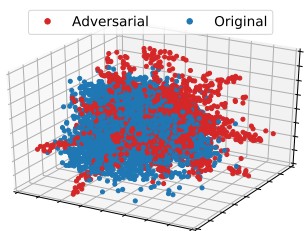

(a) SSD, a SOTA deep learning based anomaly detection approach is unable to detect a large majority of our attack samples (Here, Injected → Inj. and Detected → Det.).

(b) Visualizing the PCA components of original samples (blue) and their adversarial counterparts (red) computed for the SPICE model.

Figure 5: Results on using anomaly detection approaches as a possible defense.

**Deep Learning Based Anomaly Detection.** We now present results for SSD, the deep learning based anomaly detection approach, as a pre-processing defense to our attack. In our experiments, we start with a test set of 8000 images for all of our datasets. For each benign image $X_i$, we uniformly randomly either add adversarial noise to it or let it remain benign. We do this over 100 trials and average our results to observe how many adversarial samples were detected by the anomaly detection approach. We present these results in Fig 5a. We find that on average the method detects 11.69%, 19.41%, and 11.71%, of adversarial samples for the CIFAR-10, CIFAR-100, and STL-10 datasets, respectively. However, this is negligible compared to the large number of adversarial samples being injected and remaining undetected. To analyze the reasons for this occurrence, we compute the first 3 principal components of the adversarial set and the benign set using PCA and present these in Fig 5b for SPICE. Visually, the principal components of a large majority of adversarial samples are superimposed and interspersed amongst those of the benign samples, showcasing the potency of our attack.

## 6.3   Attacking `Face++` (MLaaS API)

We now present our results for attacking `Face++`, a production-level face clustering API service. Since the service only works with face images, we use the Extended Yale Face B [58] dataset for these experiments. There are 28 persons in this dataset, i.e., $k = 28$ and 500 images for each person. However, due to rate limits and the latency associated with uploading images using REST APIs, we test the service by sending only 10 images per person, for a total of 280 images. Then, as only cluster labels are outputted by the API, we resort to using a *surrogate model* for the attack via transferability. We train a CC model as the surrogate on the entire Extended

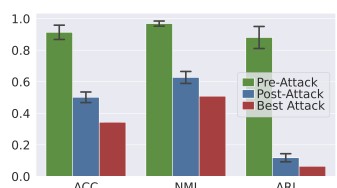

Figure 6: NMI/ACC/ARI before and after the attack on `Face++` API.

Yale Face B dataset and then train our GAN network to generate the adversarial noise for a given image from the dataset. Thus, we generate adversarial counterparts to the original 280 benign images and carry out the attack. Subsequently, we observe the NMI/ACC/ARI pre-attack and post-attack.

Since the results obtained can be affected by which 10 representative images were picked for each person in the test set, we randomize this selection process and take the average over 10 runs. As can be seen in Fig 6, before the attack, the `Face++` clustering API boasts stellar performance on the test set, with average NMI $\approx 0.97$. However, for the set of adversarial images, performance of the API service degrades significantly, with the NMI dropping down to $\approx 0.62$ on average. This shows that our generated adversarial examples can disrupt the working of a high-performance real-world service,

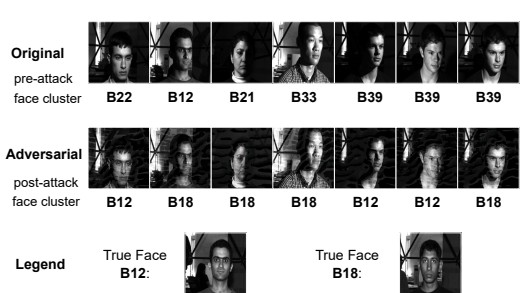

Figure 7: Adversarial samples for the `Face++` attack.

even when we cannot query the model for memberships. We also show some of the adversarial samples generated from original images, and how they have been misclustered in Fig 7. Before the attack, most of the faces are clustered correctly, but after the attack they only majorly belong to the

`B12` or `B18` face clusters. Here too, we are observing the *clustering breakdown* effect previously noted for the attack against open-source deep clustering models. We provide additional implementation details and empirical results for this attack in Appendix C.

## 7 Conclusion

In this paper, we propose the first blackbox adversarial attack against deep clustering models, using a simple GAN based architecture (Section 4.1). We evaluate our attack against a number of open-source SOTA deep clustering models and real-world image datasets, and find that it significantly reduces the performance of these models, while having minimal query complexity (Section 6.1). To further examine the robustness of deep clustering networks, we utilize two unsupervised defense approaches: adversarially trained "robust" deep clustering models, and deep learning based anomaly detection. We note that even these approaches are unable to detect our adversarial samples and mitigate the attack (Section 6.2). Finally, we use a surrogate model to attack a production-level face clustering API `Face++`, proving that our attacks are also effective against real-world MLaaS models (Section 6.3).

**Broader Impact.** Our findings are important as they underscore the need to making deep clustering models robust to real-world blackbox adversaries. As we show, current robustness/defense approaches are unsatisfactory, and allow an informed adversary to reduce the performance of models nonetheless. Another takeaway is that robust deep clustering model designers should evaluate their models against adversarial samples crafted specifically for deep clustering models (and not those crafted for supervised models, as in [9]). In the wrong hands our attack approach can have negative societal impacts, but we believe through this work we can actually *drive* the development of truly robust clustering models, thus offsetting the impact of such threats. We are positive that improved defenses and robust models will arise as a result of this work. Finally, to further detail the possible impact of our adversarial attacks, we provide practical attack scenarios in Appendix H that utilize our attack setting and threat model.

**Limitations.** There are a few limitations of our work as well: 1) For the most effective attack, all the conditions for the threat model need to be met. However, as we show for the `Face++` API, even when some conditions are not met (for `Face++` softmax cluster memberships could not be obtained), our attack can be successful. 2) Parameterization of the GAN for the attack is a non-trivial problem (for example, choosing $\epsilon$). This limits its applicability as it might be challenging to attack a model if resources are limited and efficient hyperparameter search cannot be performed. 3) Our attack assumes knowledge of the training dataset used for the model, which might not be realizable in a number of attack scenarios. To overcome this, as part of future work, we believe *partial information* attacks can be explored against deep clustering models. 4) Our attack requires training a GAN, a deep learning model, which is a computationally expensive task. It might be worthwhile to explore better optimization strategies that can utilize our $\mathcal{L}_{\text{attack}}$ loss to disrupt the deep clustering function while using less computational resources.

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
