# Appendix

**Anshuman Chhabra**[*]**, Ashwin Sekhari**[*]**, and Prasant Mohapatra**
Department of Computer Science
University of California
Davis, CA 95616
{chhabra,asekhari,pmohapatra}@ucdavis.edu

## Contents

---

[*]Equal contribution.

36th Conference on Neural Information Processing Systems (NeurIPS 2022).

# A  Additional Experiments and Results for Attack Against Deep Clustering Models

In this section, we first provide model parameters used for training the attack GANs. We provide hyperparameter details for all models (CC [1], MiCE [2], NNM [3], SCAN [4], SPICE [5], RUC [6]) and all datasets (CIFAR-10 [7], CIFAR-100 [8], and STL-10 [9]). That is, we list values of $\alpha_a, \alpha_c, \epsilon$, and clamping for all these models. Note that all the aforementioned deep clustering models are trained as described by the authors (either in their papers or Github repositories). We also seek to utilize pre-trained models for these wherever provided.

We then provide sample images from each cluster/class for each of the models, along with the generated noise using our GAN models. We also show the original and predicted cluster labels to show how the attack leads to misclustering.

Next, we provide additional confusion matrices (other than CC and SPICE as in the main paper) for all the 3 datasets considered. These provide additional insight into the clustering breakdown effect, and how the clustering output changes for each model.

Then we provide additional plots detailing performance (NMI/ACC/ARI) vs the perturbation norm ($\epsilon$) for all the datasets and performance metrics. All the trends are very similar, but are useful indicators for how much noise is needed to degrade performance significantly.

After these plots, we provide the remaining transferability matrices for all datasets and performance metrics (NMI/ACC/ARI). In the subsequent subsection, we list analytical definitions for all the performance metrics considered in the paper.

## A.1  Model Parameters and Implementation Details

All the code for the deep clustering models and our GAN models is written in PyTorch[2].We consider the same training/test split as considered by authors in their papers while evaluating models. We thus train our GAN to attack the model as well as evaluate the attack by considering the split in a similar fashion.

We now present the hyperparameters used for training all the GAN attack models against each of the open-source deep clustering models (Tables 1 - 5).

### A.1.1  CC

| | $\alpha_a$ | $\alpha_c$ | $\epsilon$ | clamping |
|---|---|---|---|---|
| CIFAR-10 | 5 | 1 | 7.95 | 0.03 |
| CIFAR-100 | 5 | 1 | 10.08 | 0.05 |
| STL-10 | 5 | 1 | 10.11 | 0.03 |

Table 1: Implementation parameters for CC

### A.1.2  MICE

| | $\alpha_a$ | $\alpha_c$ | $\epsilon$ | clamping |
|---|---|---|---|---|
| CIFAR-10 | 30 | 1 | 5.54 | 0.1 |
| CIFAR-100 | 30 | 1 | 5.50 | 0.1 |
| STL-10 | 30 | 1 | 23.99 | 0.15 |

Table 2: Implementation parameters for MICE

---

[2]https://pytorch.org/

### A.1.3 NNM

| | $\alpha_a$ | $\alpha_c$ | $\epsilon$ | clamping |
|---|---|---|---|---|
| CIFAR-10 | 30 | 1 | 15.37 | 0.3 |
| CIFAR-100 | 30 | 1 | 5.41 | 0.1 |
| STL-10 | 30 | 1 | 32.37 | 0.2 |

Table 3: Implementation parameters for NNM

### A.1.4 SCAN

| | $\alpha_a$ | $\alpha_c$ | $\epsilon$ | clamping |
|---|---|---|---|---|
| CIFAR-10 | 30 | 1 | 10.87 | 0.2 |
| CIFAR-100 | 30 | 1 | 8.23 | 0.15 |
| STL-10 | 30 | 1 | 19.55 | 0.12 |

Table 4: Implementation parameters for SCAN

### A.1.5 SPICE

| | $\alpha_a$ | $\alpha_c$ | $\epsilon$ | clamping |
|---|---|---|---|---|
| CIFAR-10 | 10 | 1 | 12.04 | 0.22 |
| CIFAR-100 | 10 | 1 | 9.29 | 0.17 |
| STL-10 | 10 | 1 | 104.04 | 0.2 |

Table 5: Implementation parameters for SPICE

## A.2 Visualizing Cluster-wise Generated Adversarial Images

We now present some sample adversarial images for all models on all the datasets. Majority cluster labels for each image shown are depicted right above the image. We also show the adversarial noise crafted by our generators for each specific sample. Refer to Figures 1 - 5 for CIFAR-10, Figures 6 - 10 for CIFAR-100, and Figures 11 - 15 for STL-10.

### A.2.1 CIFAR-10

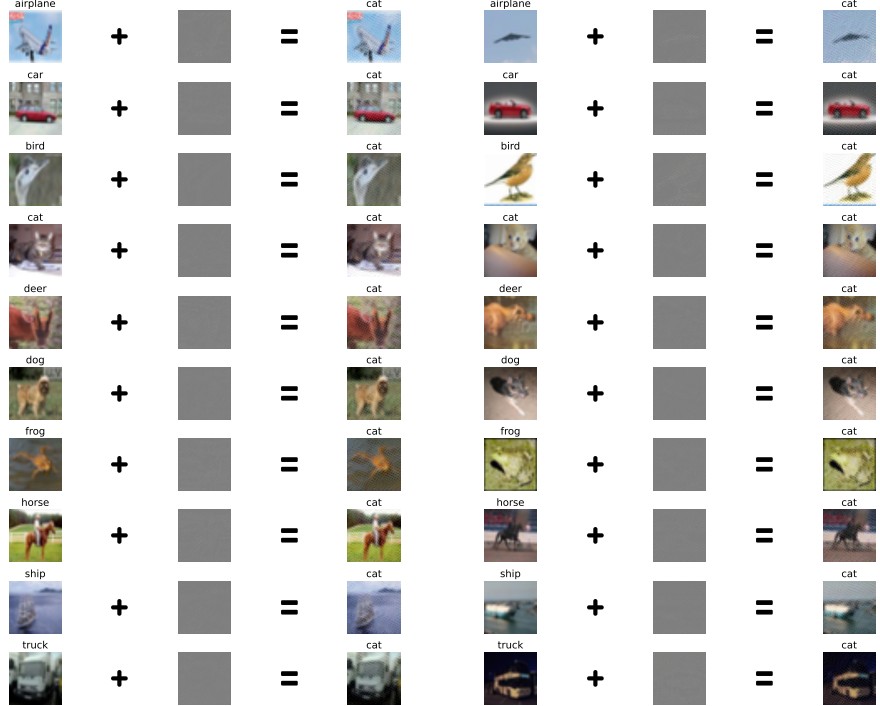

Figure 1: CC (CIFAR-10)

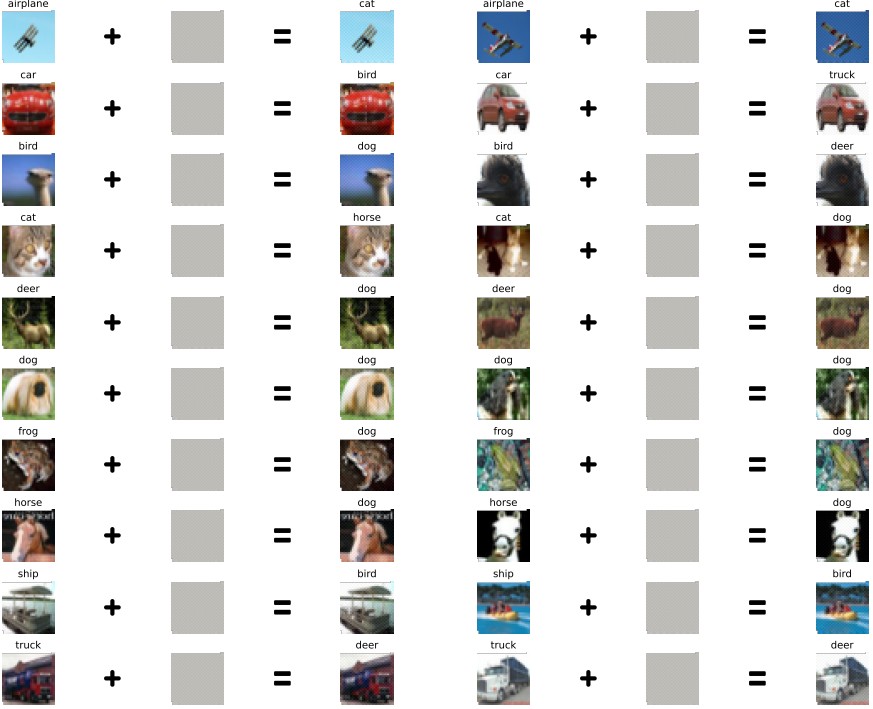

Figure 2: MiCE (CIFAR-10)

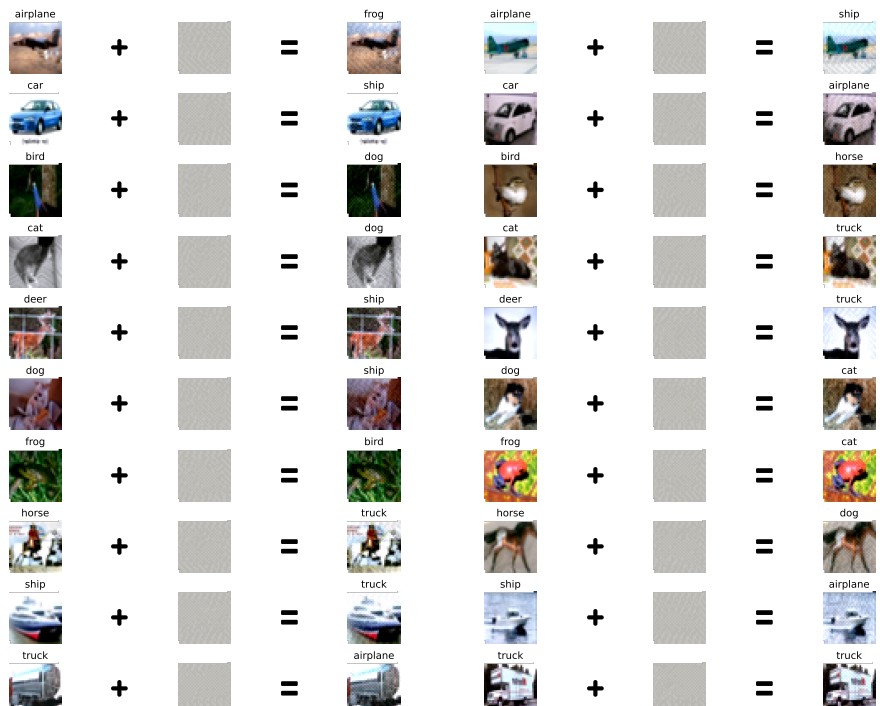

Figure 3: NNM (CIFAR-10)

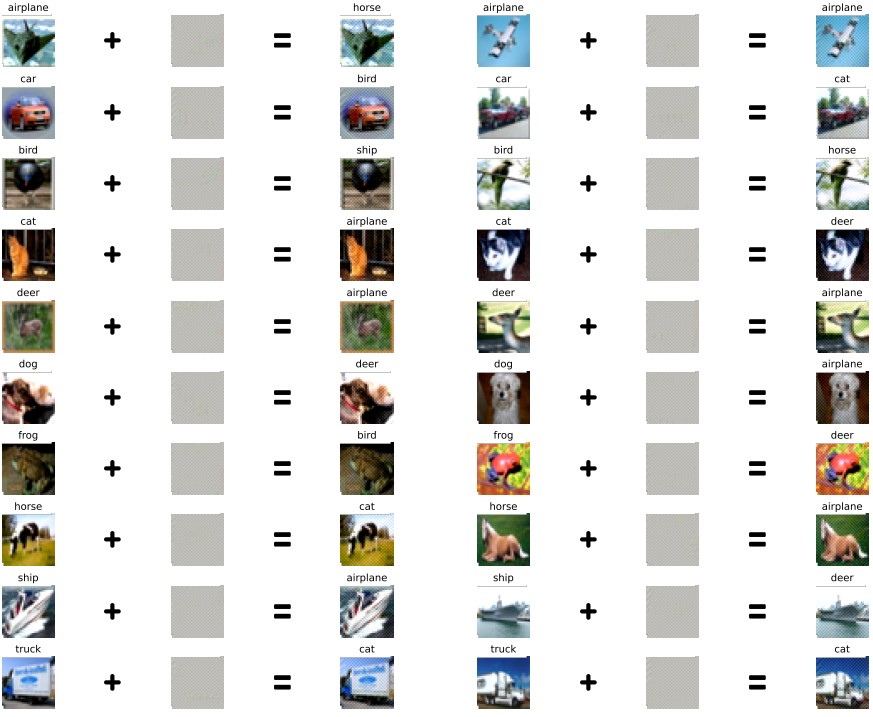

Figure 4: SCAN (CIFAR-10)

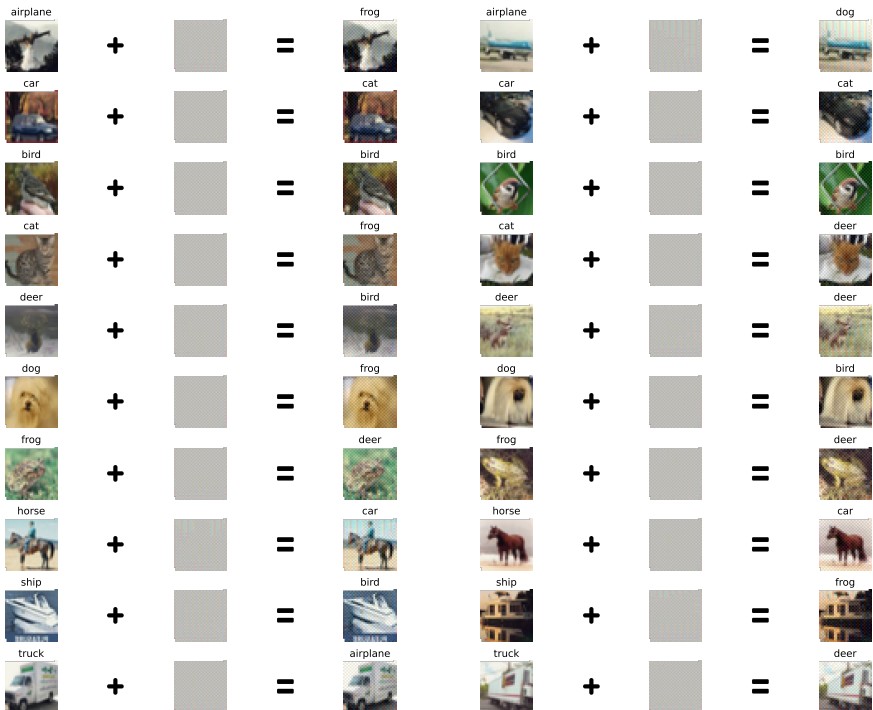

Figure 5: SPICE (CIFAR-10)

## A.2.2 CIFAR-100

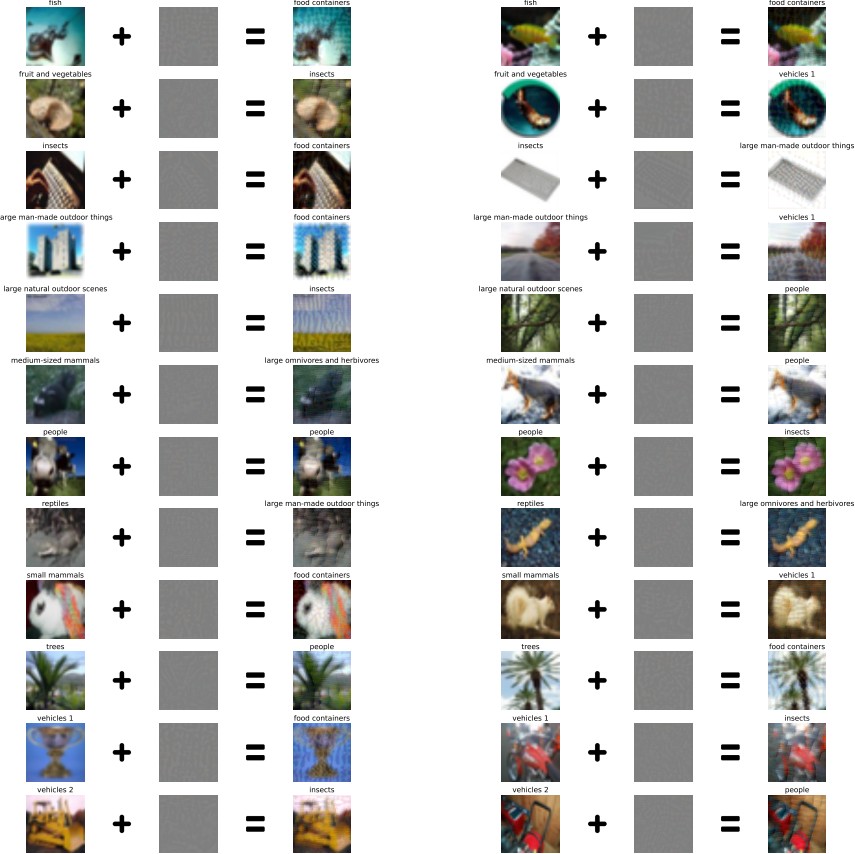

Figure 6: CC (CIFAR-100)

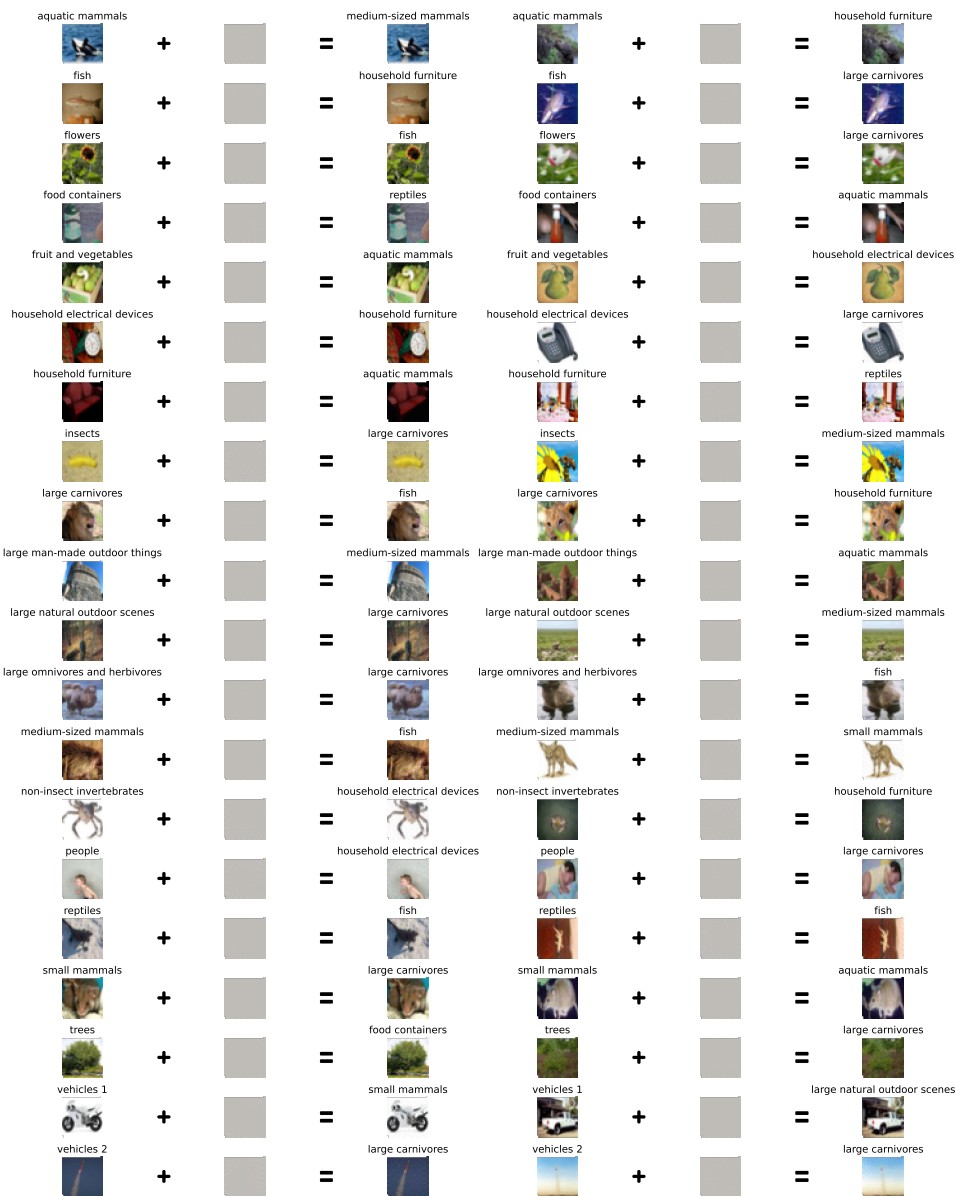

Figure 7: MiCE (CIFAR-100)

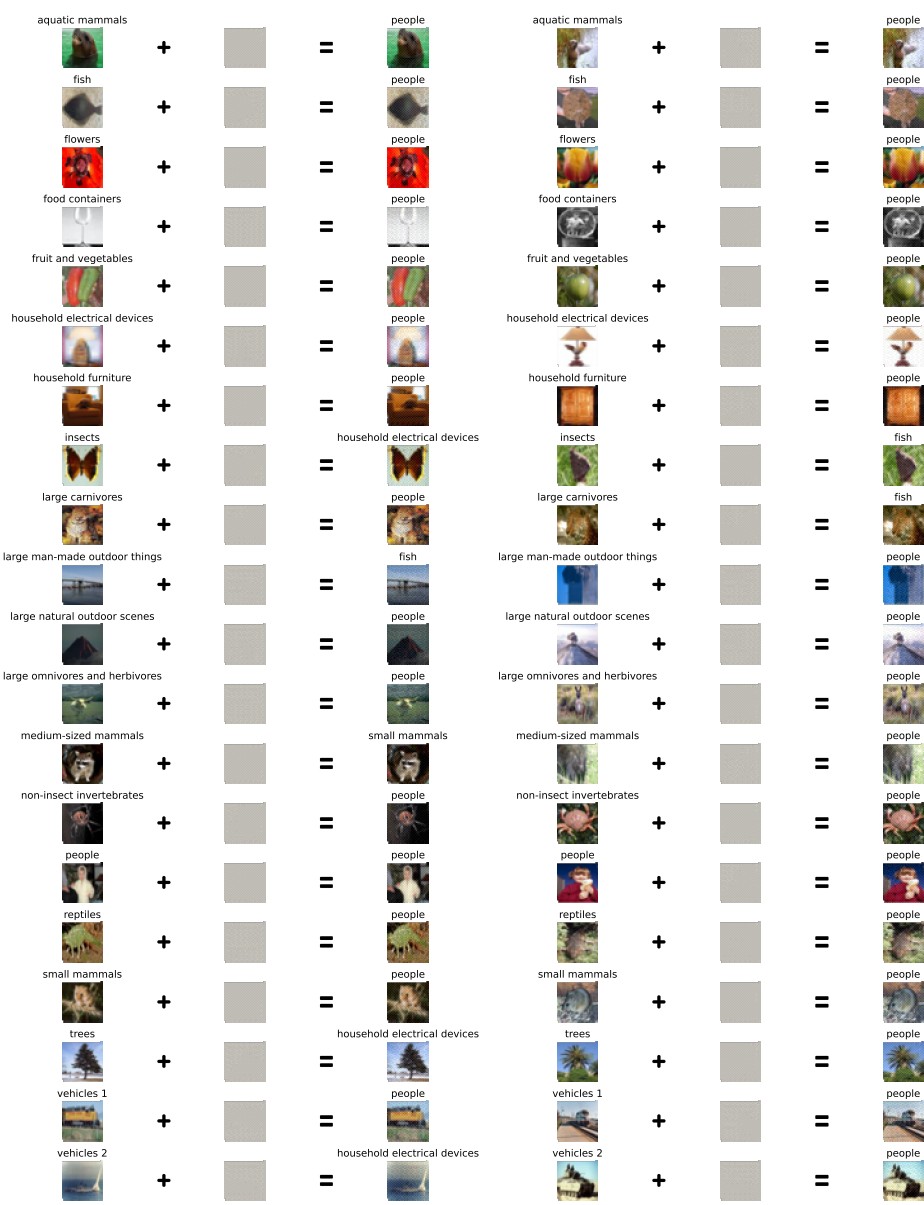

Figure 8: NNM (CIFAR-100)

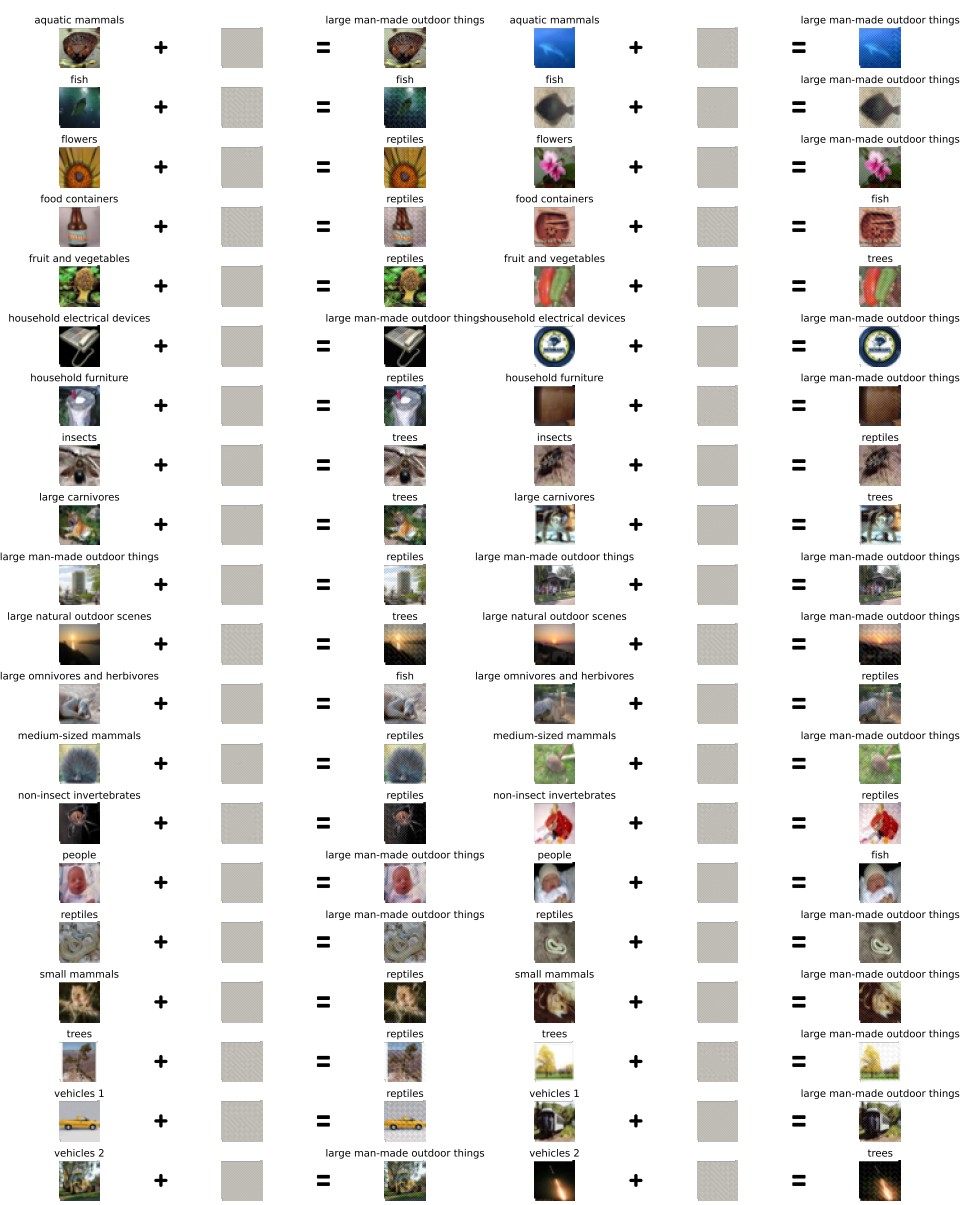

Figure 9: SCAN (CIFAR-100)

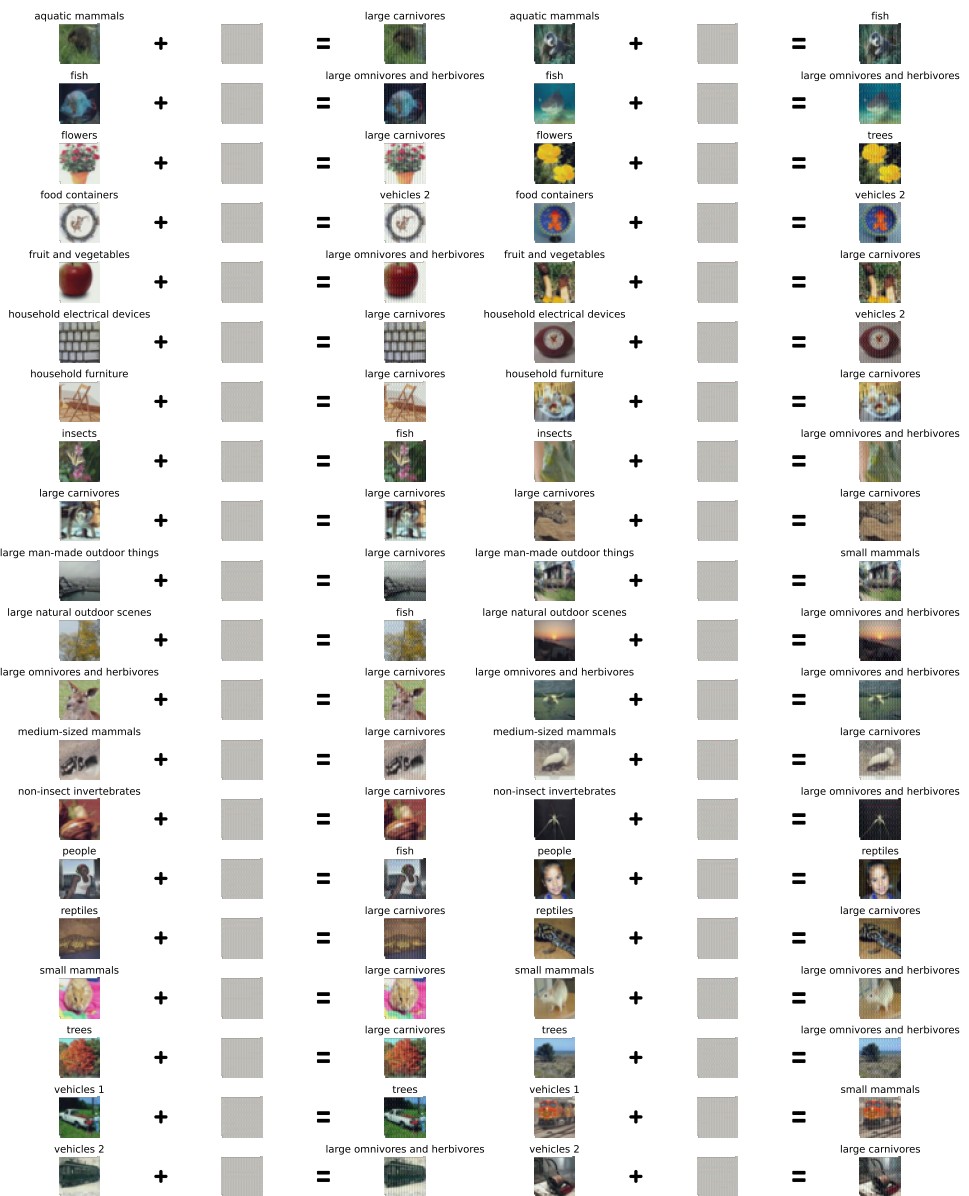

Figure 10: SPICE (CIFAR-100)

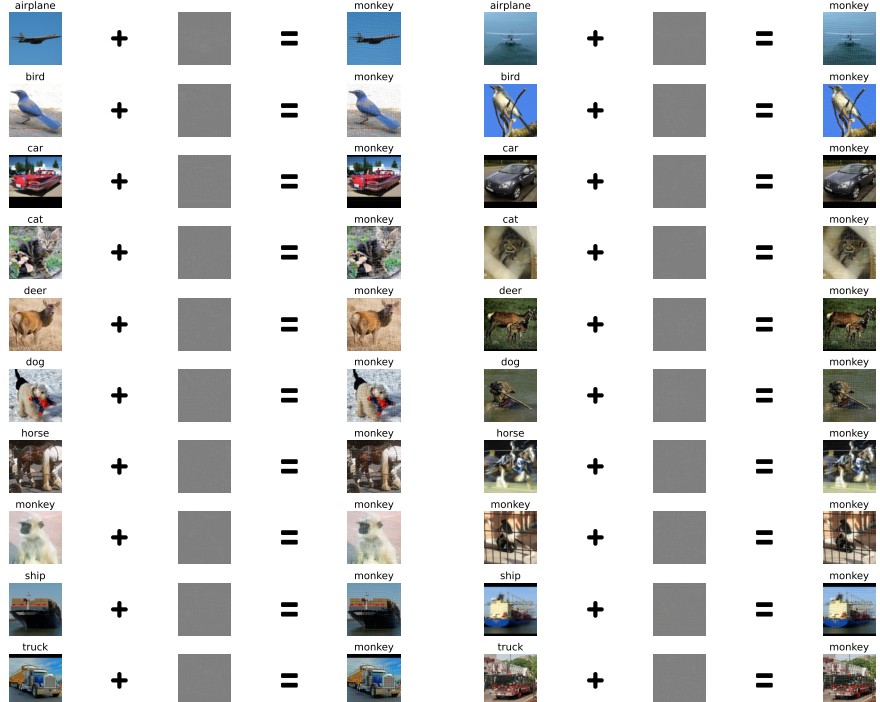

Figure 11: CC (STL-10)

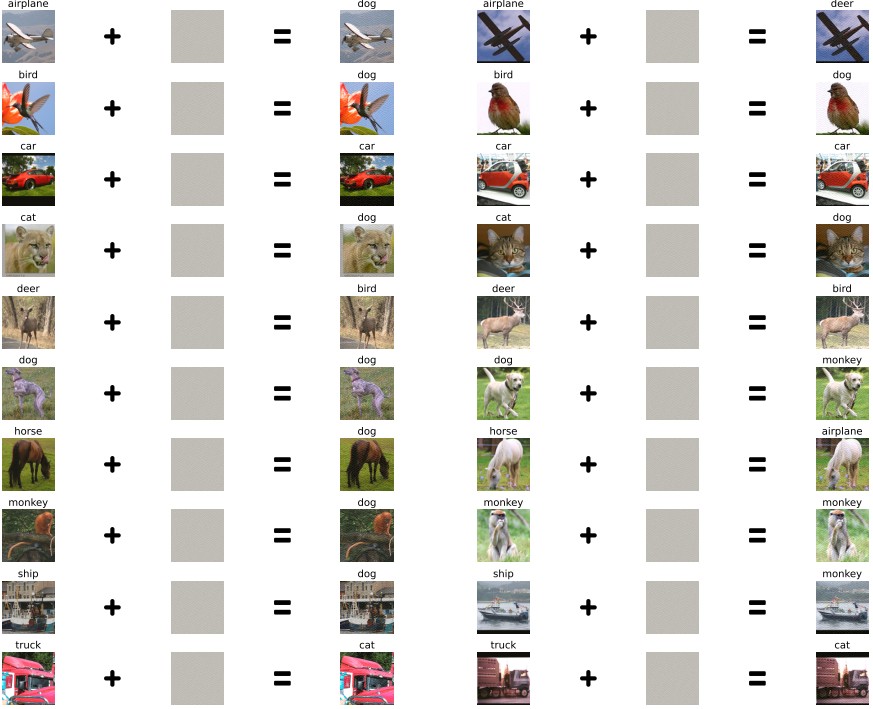

Figure 12: MiCE (STL-10)

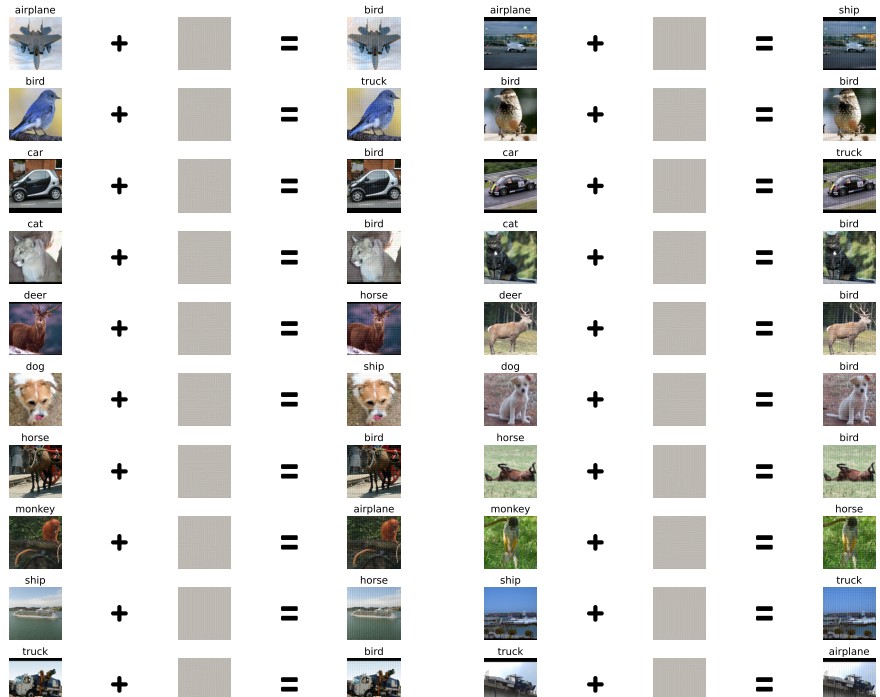

Figure 13: NNM (STL-10)

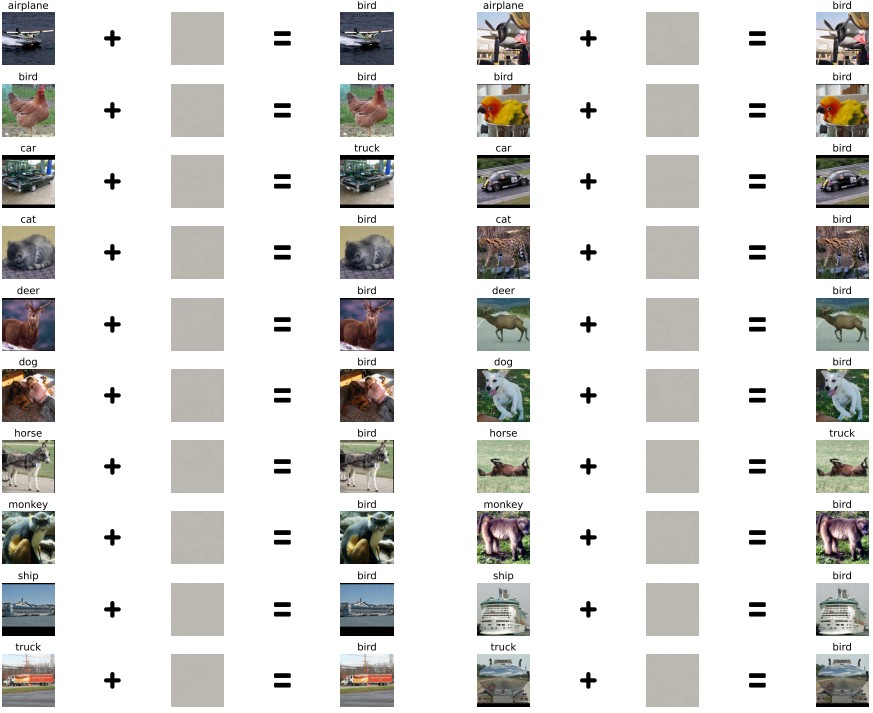

Figure 14: SCAN (STL-10)

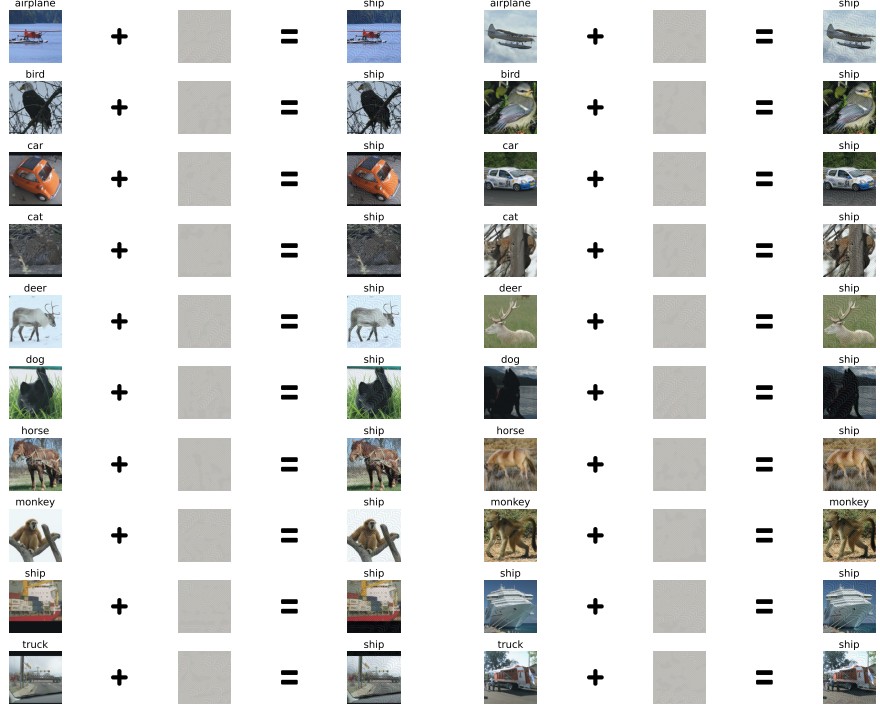

Figure 15: SPICE (STL-10)

## A.3 Additional Confusion Matrices

We now present additional confusion matrices for the datasets and deep clustering models showcasing the clustering before and after the attack. We show the pre-attack clustering outputs, the confusion matrices for results reported in the paper, and a *best* attack scenario where we show the most potent and degrading attack possible using our GAN attack. Refer to Figures 16 - 20 for CIFAR-10, Figures 21 - 25 for CIFAR-100, and Figures 26 - 30 for STL-10.

### A.3.1 CIFAR-10

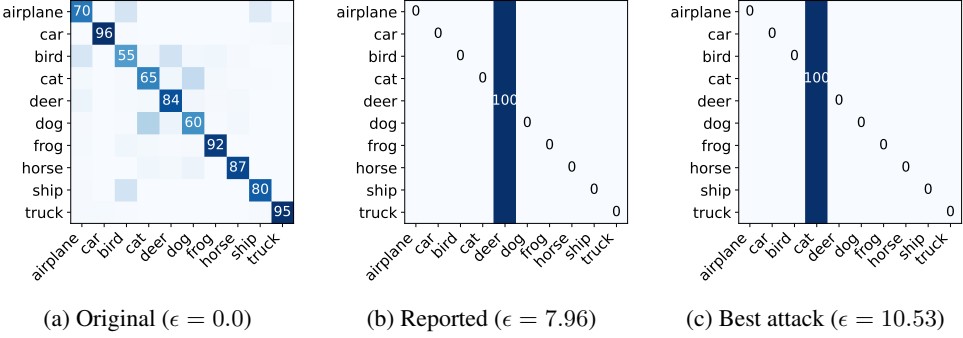

(a) Original ($\epsilon = 0.0$)      (b) Reported ($\epsilon = 7.96$)      (c) Best attack ($\epsilon = 10.53$)

Figure 16: Confusion Matrices for CC (CIFAR-10)

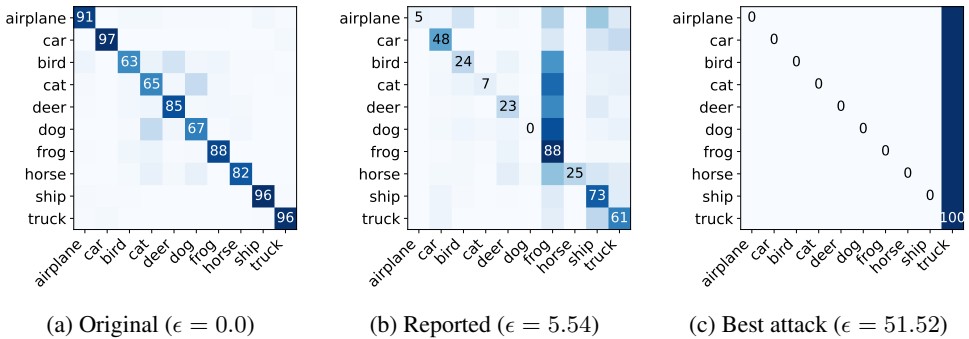

(a) Original ($\epsilon = 0.0$)      (b) Reported ($\epsilon = 5.54$)      (c) Best attack ($\epsilon = 51.52$)

Figure 17: Confusion Matrices for MiCE (CIFAR-10)

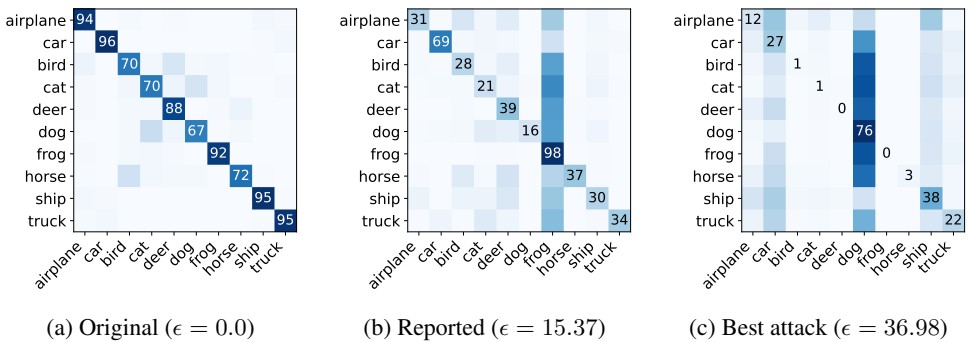

(a) Original ($\epsilon = 0.0$)      (b) Reported ($\epsilon = 15.37$)      (c) Best attack ($\epsilon = 36.98$)

Figure 18: Confusion Matrices for NNM (CIFAR-10)

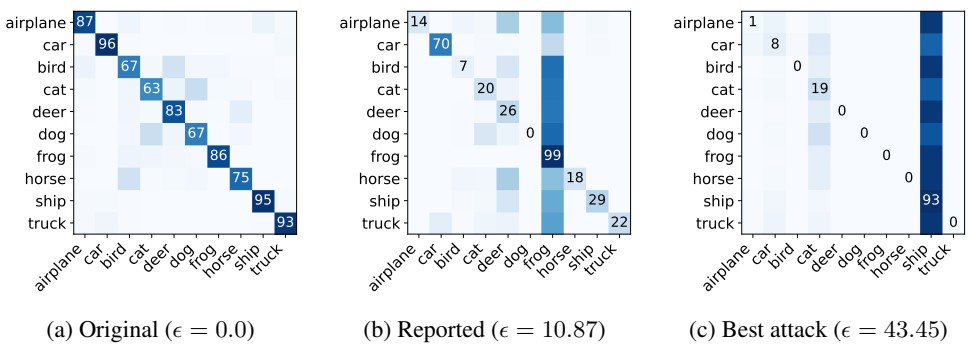

(a) Original ($\epsilon = 0.0$)      (b) Reported ($\epsilon = 10.87$)      (c) Best attack ($\epsilon = 43.45$)

Figure 19: Confusion Matrices for SCAN (CIFAR-10)

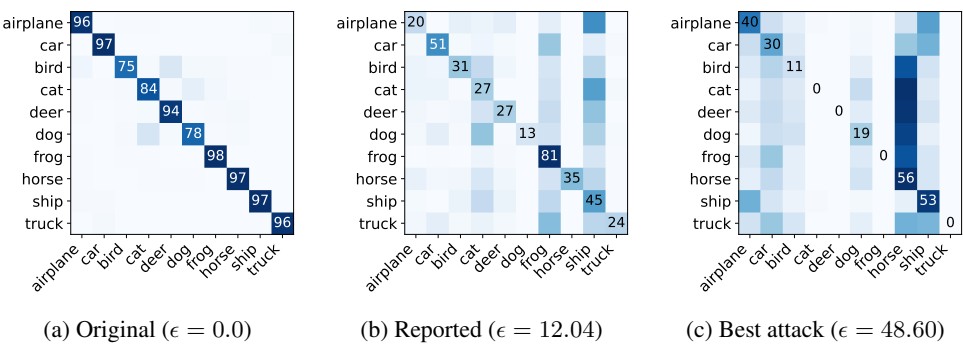

(a) Original ($\epsilon = 0.0$)      (b) Reported ($\epsilon = 12.04$)      (c) Best attack ($\epsilon = 48.60$)

Figure 20: Confusion Matrices for SPICE (CIFAR-10)

### A.3.2 CIFAR-100

For ease of visualization, in these experiments we condense CIFAR-100 into CIFAR-20. CIFAR-100 contains 100 classes which can be represented by 20 superlabels, which constitute CIFAR-20. Furthermore to improve readability, we denote each of these superlabels using the following labels: {A, B, C, D, E, F, G, H, I, J, K, L, M, N, O, P, Q, R, S, and T}. These labels are a one-to-one direct mapping of the original superlabels in order: {"aquatic mammals", "fish", "flowers","food containers","fruit and vegetables","household electrical devices","household furniture","insects","large carnivores","large man-made outdoor things","large natural outdoor scenes","large omnivores and herbivores","medium-sized mammals","non-insect invertebrates","people","reptiles", "small mammals","trees","vehicles 1", and "vehicles 2"}.

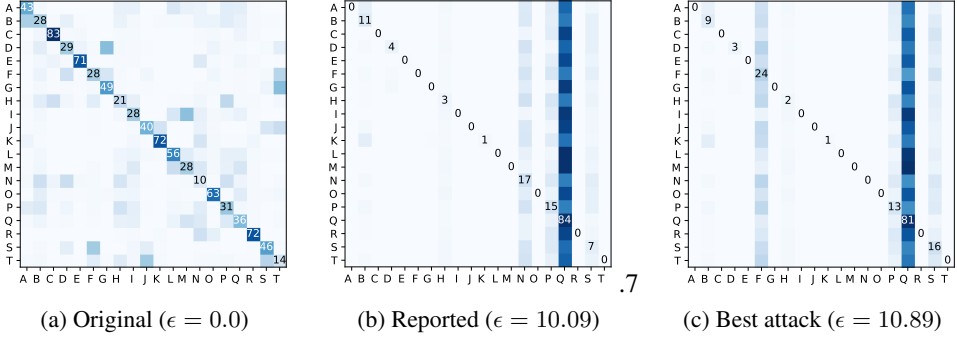

(a) Original ($\epsilon = 0.0$)  (b) Reported ($\epsilon = 10.09$)  (c) Best attack ($\epsilon = 10.89$)

Figure 21: Confusion Matrices for CC (CIFAR-100)

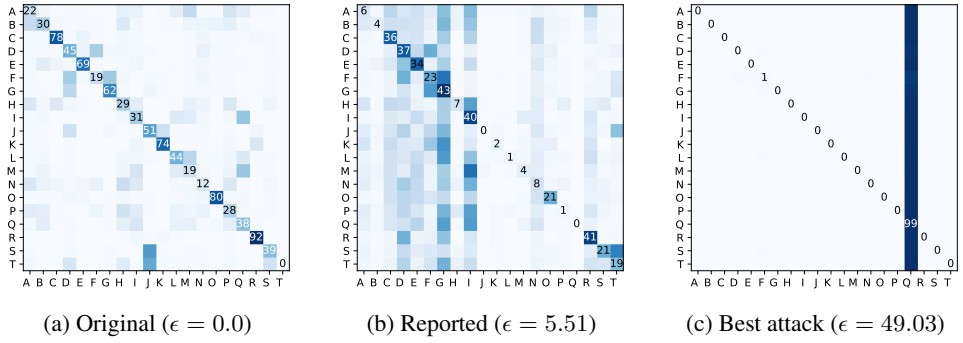

(a) Original ($\epsilon = 0.0$)  (b) Reported ($\epsilon = 5.51$)  (c) Best attack ($\epsilon = 49.03$)

Figure 22: Confusion Matrices for MiCE (CIFAR-100)

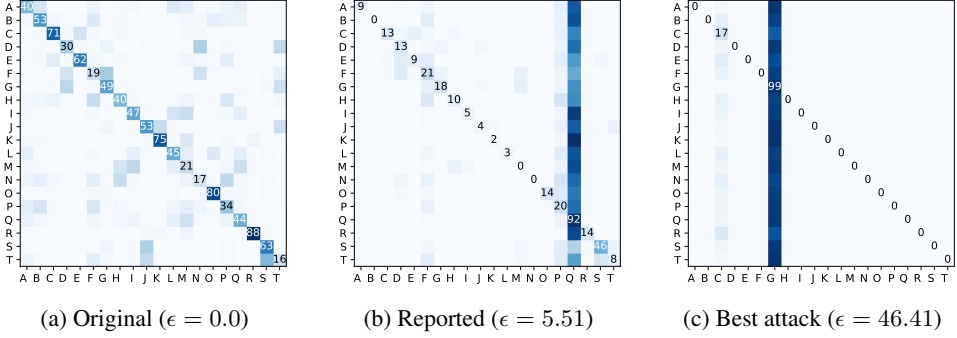

(a) Original ($\epsilon = 0.0$)  (b) Reported ($\epsilon = 5.51$)  (c) Best attack ($\epsilon = 46.41$)

Figure 23: Confusion Matrices for NNM (CIFAR-100)

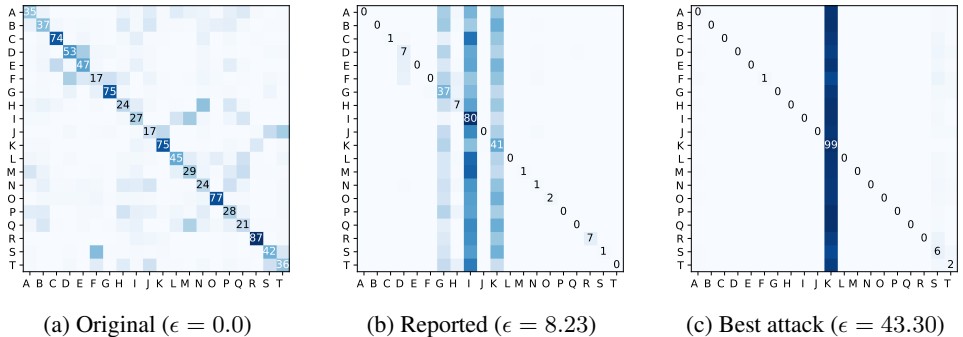

(a) Original ($\epsilon = 0.0$)    (b) Reported ($\epsilon = 8.23$)    (c) Best attack ($\epsilon = 43.30$)

Figure 24: Confusion Matrices for SCAN (CIFAR-100)

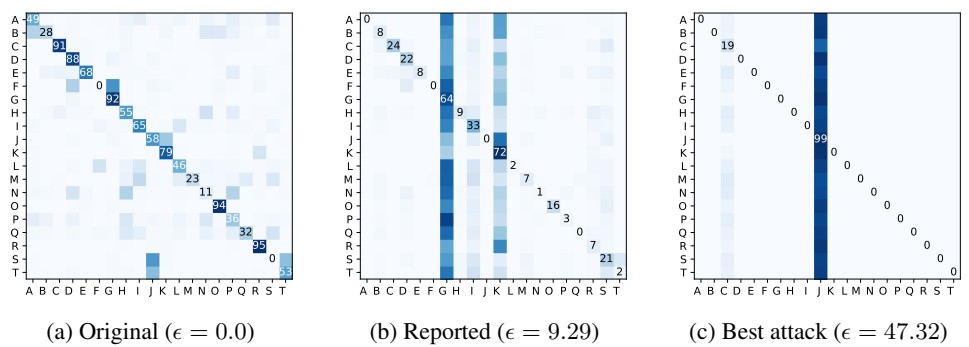

(a) Original ($\epsilon = 0.0$)    (b) Reported ($\epsilon = 9.29$)    (c) Best attack ($\epsilon = 47.32$)

Figure 25: Confusion Matrices for SPICE (CIFAR-100)

### A.3.3 STL-10

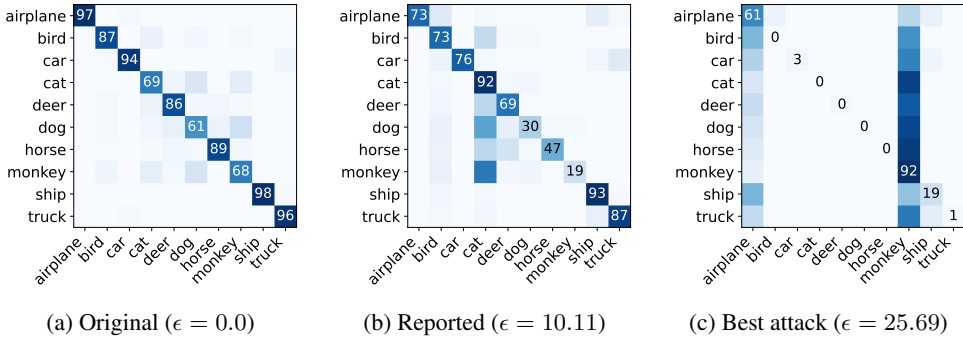

(a) Original ($\epsilon = 0.0$)    (b) Reported ($\epsilon = 10.11$)    (c) Best attack ($\epsilon = 25.69$)

Figure 26: Confusion Matrices for CC (STL-10)

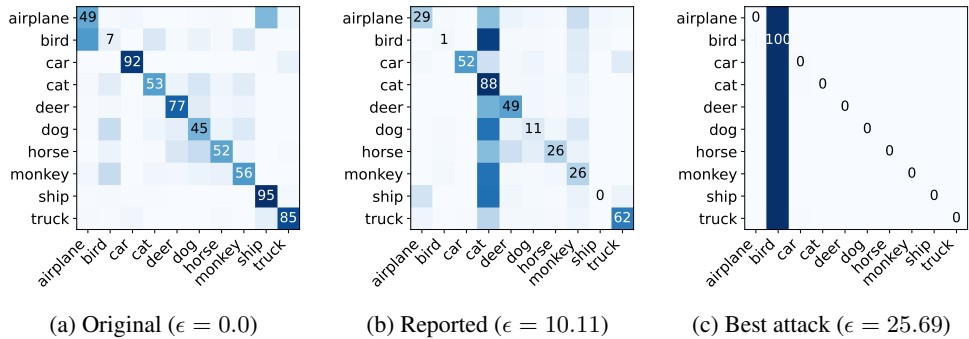

(a) Original ($\epsilon = 0.0$)  (b) Reported ($\epsilon = 10.11$)  (c) Best attack ($\epsilon = 25.69$)

Figure 27: Confusion Matrices for MiCE (STL-10)

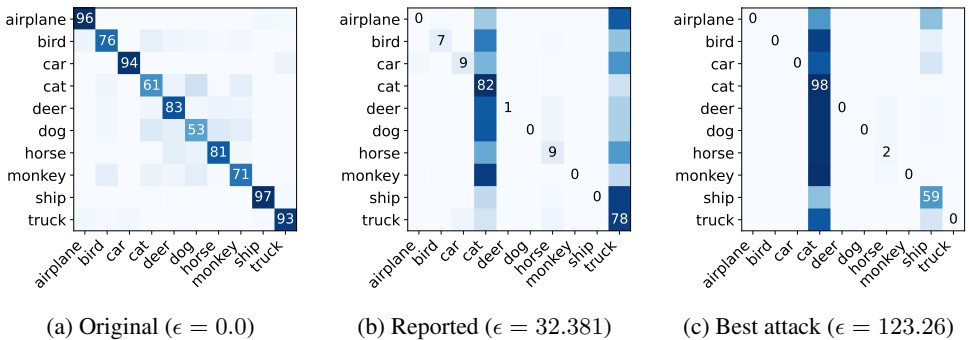

(a) Original ($\epsilon = 0.0$)  (b) Reported ($\epsilon = 32.381$)  (c) Best attack ($\epsilon = 123.26$)

Figure 28: Confusion Matrices for NNM (STL-10)

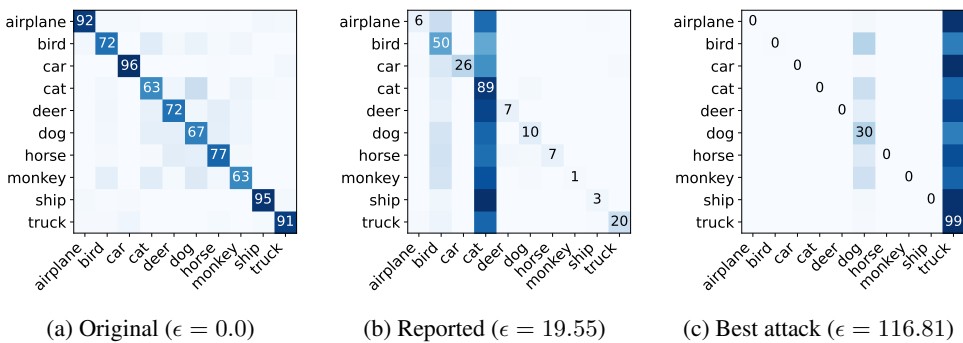

(a) Original ($\epsilon = 0.0$)  (b) Reported ($\epsilon = 19.55$)  (c) Best attack ($\epsilon = 116.81$)

Figure 29: Confusion Matrices for SCAN (STL-10)

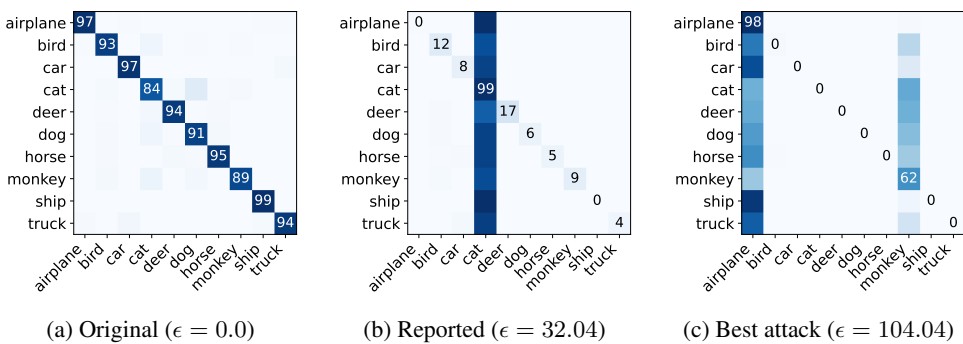

(a) Original ($\epsilon = 0.0$)  (b) Reported ($\epsilon = 32.04$)  (c) Best attack ($\epsilon = 104.04$)

Figure 30: Confusion Matrices for SPICE (STL-10)

## A.4 Additional Performance vs Perturbation Norm Plots

We now present additional performance versus the norm of the crafted perturbation plots (Figure 31 for CIFAR-10, Figure 32 for CIFAR-100, and Figure 33 for STL-10).

### A.4.1 CIFAR-10

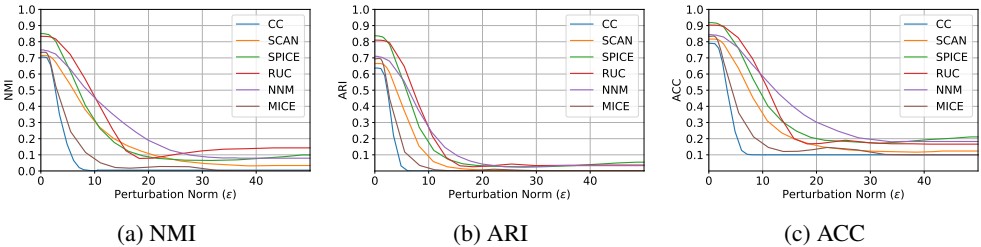

(a) NMI        (b) ARI        (c) ACC

Figure 31: Performance v.s. Perturbation Norm plots for CIFAR-10

### A.4.2 CIFAR-100

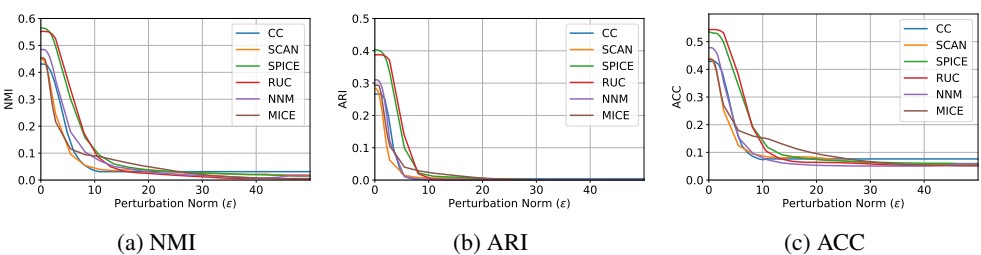

(a) NMI        (b) ARI        (c) ACC

Figure 32: Performance v.s. Perturbation Norm plots for CIFAR-100

### A.4.3 STL-10

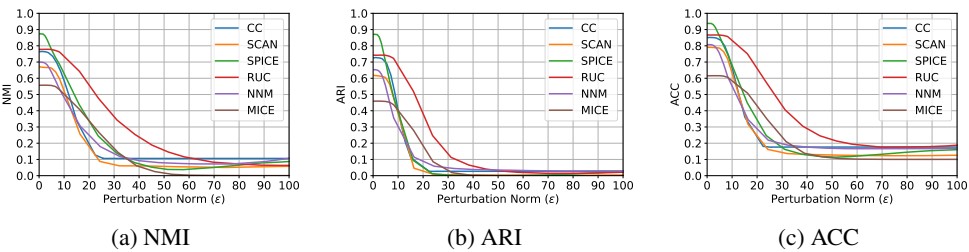

(a) NMI        (b) ARI        (c) ACC

Figure 33: Performance v.s. Perturbation Norm plots for STL-10

## A.5 Additional Transferability Matrices

Here we present the additional transferability analysis which shows ARI and ACC as well. Note that RUC is also included, for ease of comparison but other plots for RUC are shown in later sections. Refer to Figure 34 for CIFAR-10 matrices, Figure 35 for CIFAR-100, and Figure 36 for STL-10.

### A.5.1 CIFAR-10

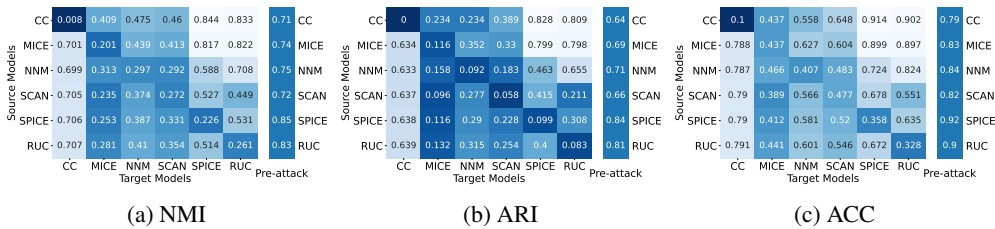

(a) NMI        (b) ARI        (c) ACC

Figure 34: Transferability matrices for CIFAR-10

### A.5.2 CIFAR-100

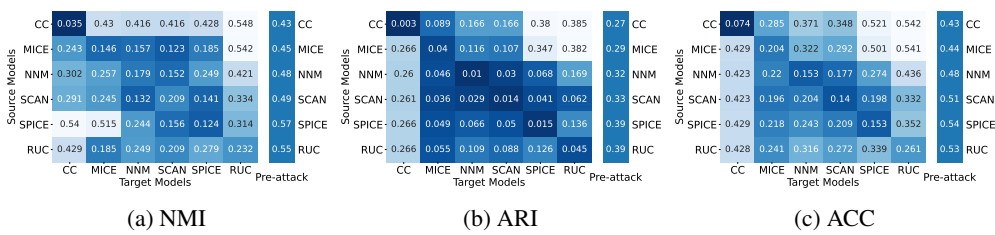

(a) NMI        (b) ARI        (c) ACC

Figure 35: Transferability matrices for CIFAR-100

### A.5.3 STL-10

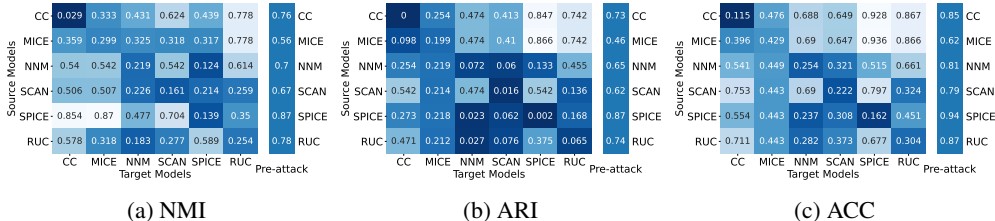

(a) NMI        (b) ARI        (c) ACC

Figure 36: Transferability matrices for STL-10

## A.6 Definitions for Metrics

We have considered the following 3 performance metrics in the paper: Unsupervised Accuracy (ACC) [10], Normalized Mutual Information (NMI) [11], and Adjusted Rand Index (ARI) [12]. We provide analytical definitions for these below:

**ACC.** This is the unsupervised equivalent of the traditional classification accuracy. Let there be a mapping function $\phi$ that computes all possible mappings between ground truth labels and possible cluster assignments[3] for some $m$ samples. Also let $Y_i, L_i$ denote the ground truth label and cluster assignment label for the $i$-th sample, respectively. Then we can define ACC as:

$$\text{ACC} = \max_{\phi} \frac{\sum_{i=1}^{m} \mathbb{1}\{Y_i = \phi(L_i)\}}{m}$$

**NMI.** Normalized Mutual Information is essentially a normalized version of the widely used mutual information metric. Let $I$ denote the mutual information metric [14], $E$ denote entropy [14], $L$ denote the cluster assignment labels, and $Y$ denote the ground truth labels. Then we can define NMI as:

---

[3]Such a mapping function can be computed optimally using the Hungarian assignment algorithm [13].

$$\text{NMI} = \frac{I(Y, L)}{(1/2) * [E(Y) + E(L)]}$$

**ARI.** The Adjusted Rand Index is based on the Rand Index, commonly used in statistical literature [15]. Let $R$ denote the "unadjusted" Rand Index, then we can write ARI as:

$$\text{ARI} = \frac{R - \mathbb{E}[R]}{\max(R) - \mathbb{E}[R]}$$

# B  Additional Details on Defense Approaches

In this section, we provide additional details for the defense approaches considered in this paper. First, we provide additional results for the RUC [6] robust deep clustering model, and then for SSD [16], which constitutes the pre-processing anomaly detection defense.

## B.1  Robust Deep Clustering

We provide hyperparameter values (Table 6) for training the GAN network for RUC, along with confusion matrices (Figures 37 - 39) and adversarial samples (Figures 40 - 42) obtained via our attack.

### B.1.1  Hyperparameter Values

|  | $\alpha_a$ | $\alpha_c$ | $\epsilon$ | clamping |
|---|---|---|---|---|
| CIFAR-10 | 5 | 1 | 13.28 | 0.25 |
| CIFAR-100 | 5 | 1 | 8.03 | 0.15 |
| STL-10 | 5 | 1 | 38.34 | 0.25 |

Table 6: Implementation parameters for RUC

### B.1.2  Confusion Matrices

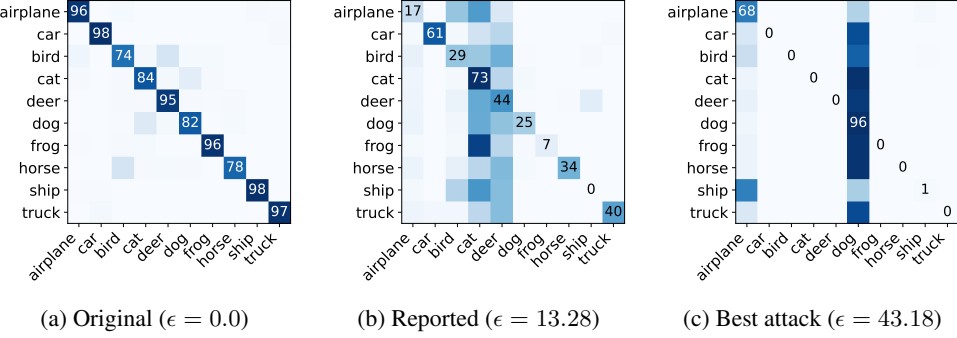

(a) Original ($\epsilon = 0.0$)  (b) Reported ($\epsilon = 13.28$)  (c) Best attack ($\epsilon = 43.18$)

Figure 37: Confusion Matrices for RUC (CIFAR-10)

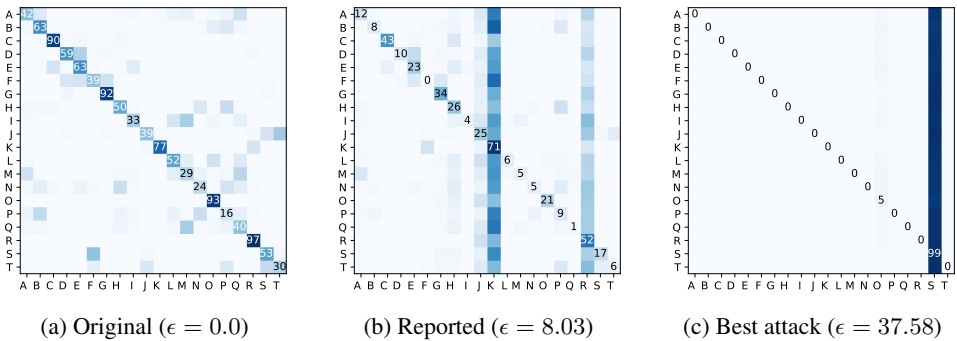

(a) Original ($\epsilon = 0.0$)    (b) Reported ($\epsilon = 8.03$)    (c) Best attack ($\epsilon = 37.58$)

Figure 38: Confusion Matrices for RUC (CIFAR-100)

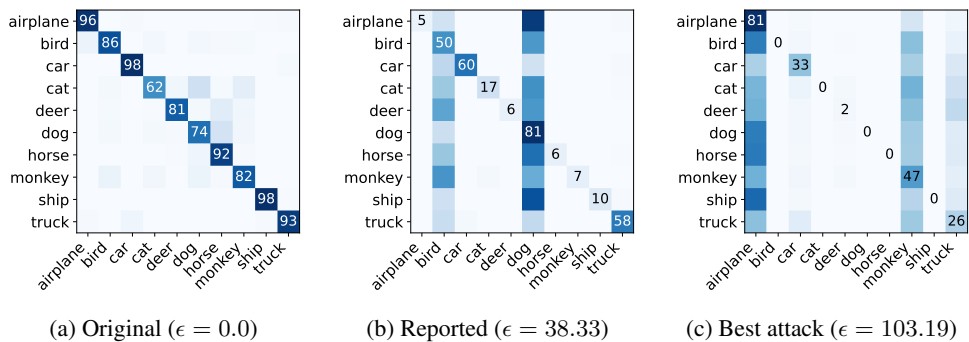

(a) Original ($\epsilon = 0.0$)    (b) Reported ($\epsilon = 38.33$)    (c) Best attack ($\epsilon = 103.19$)

Figure 39: Confusion Matrices for RUC (STL-10)

### B.1.3    Adversarial Samples for RUC

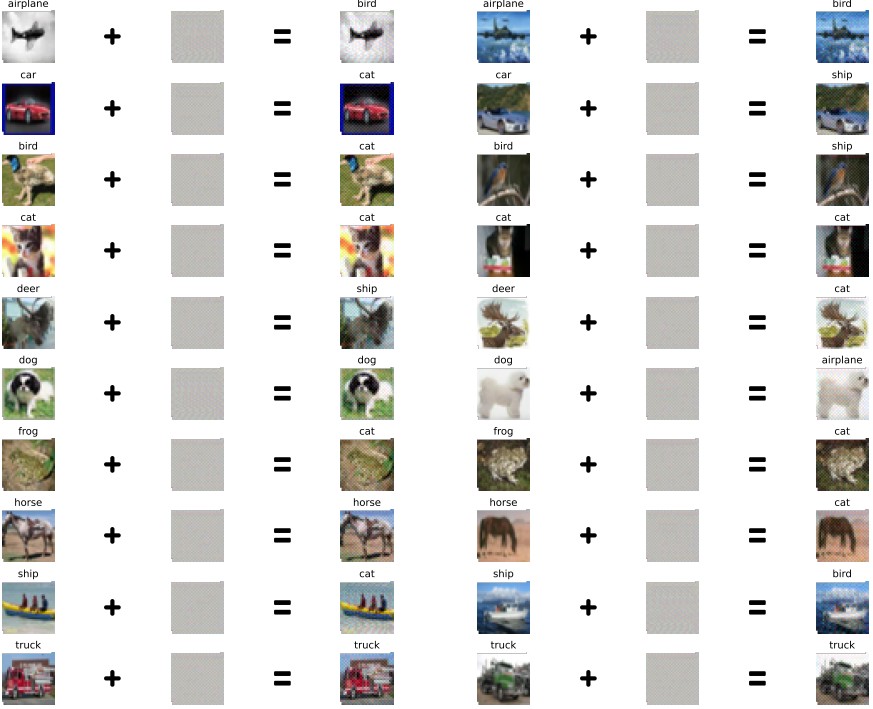

Figure 40: RUC (CIFAR-10)

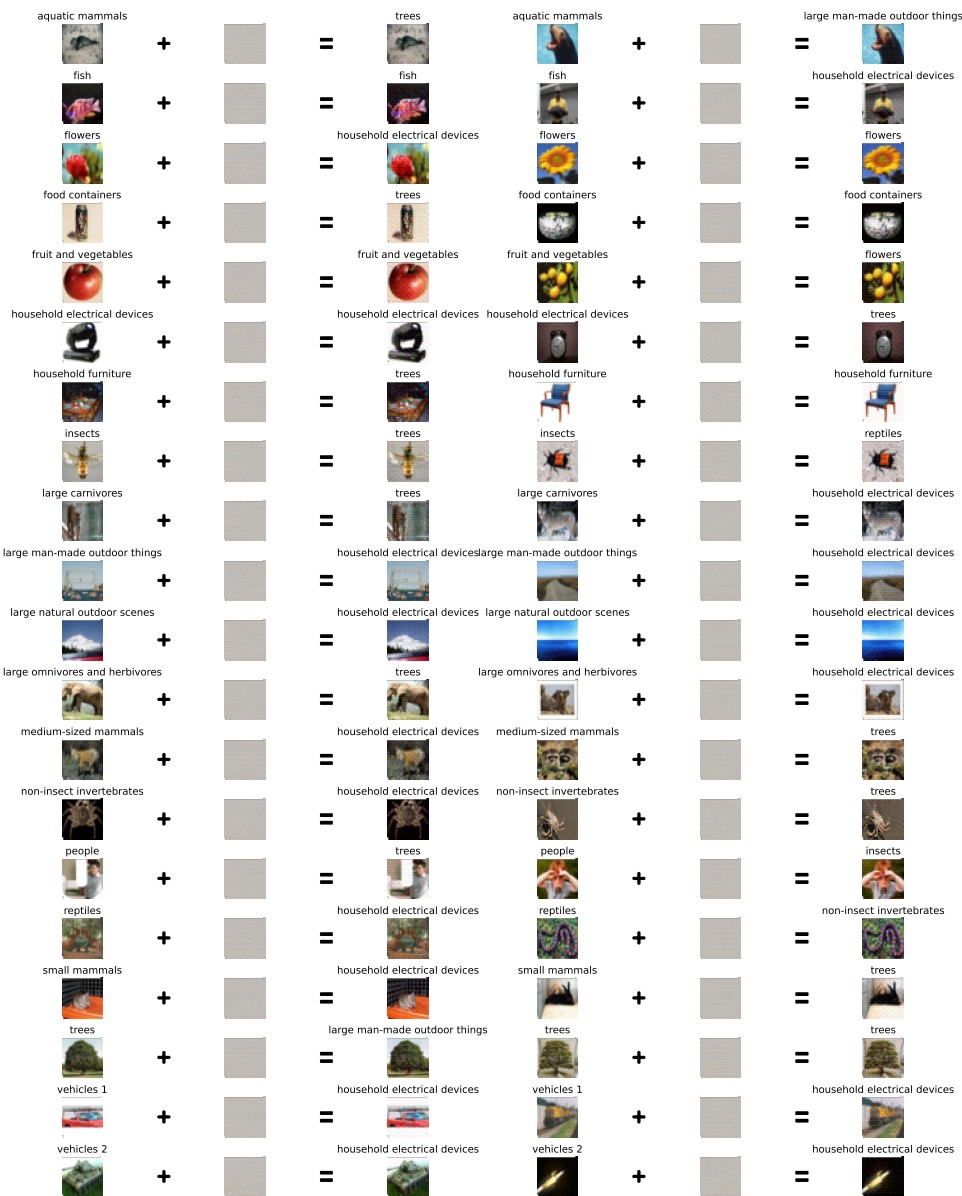

Figure 41: RUC (CIFAR-100)

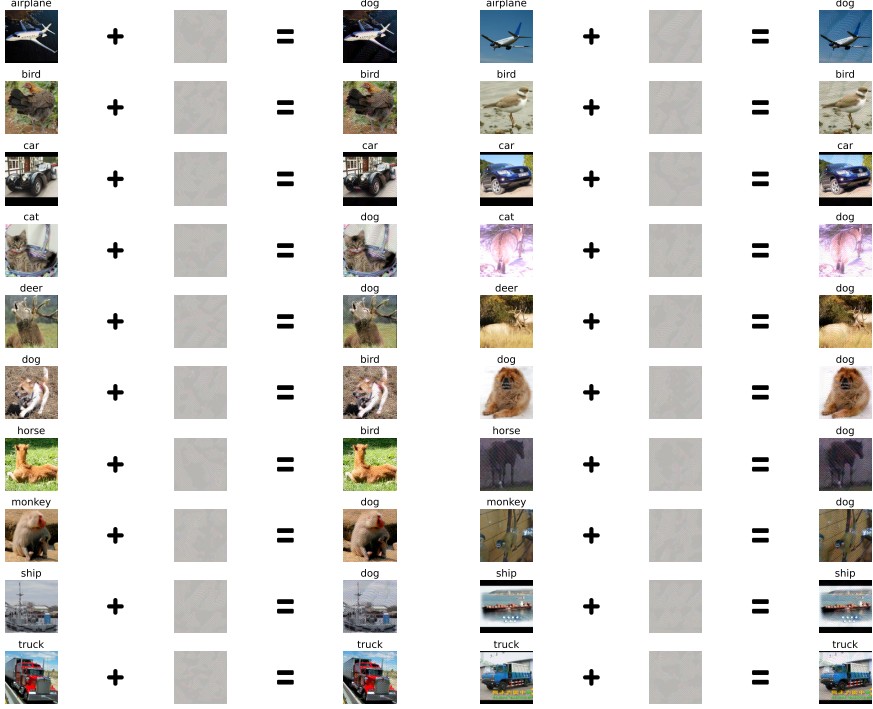

Figure 42: RUC (STL-10)

## B.2 Deep Learning Based Anomaly Detection

We now present additional details regarding the anomaly detection defense approach. The Github repository provided by the authors of SSD[4] only provides pretrained mdoels for CIFAR-10, and for the ResNet50 architecture. However our models (CC, MICE, NNM, SCAN, SPICE, RUC) use either ResNet18, or ResNet34 as the base architecture. Therefore, for each of our models, we retrain the SSD models from scratch, and provide the training hyperparameters in Table 7.

Then, in Table 8 we provide the actual values used for generating the injection/detection bar plot figure in the main text.

| Dataset | Architecture | Learning Rate | Epochs | Training Mode |
|---------|--------------|---------------|--------|---------------|
| CIFAR-10 | ResNet18 | 0.5 | 500 | SimCLR |
| CIFAR-10 | ResNet34 | 0.5 | 500 | SimCLR |
| CIfAR-100 | Resnet18 | 0.5 | 500 | SimCLR |
| CIFAR-100 | ResNet34 | 0.5 | 500 | SimCLR |
| STL-10 | Resnet18 | 0.5 | 500 | SimCLR |
| STL-10 | ResNet34 | 0.5 | 500 | SimCLR |

Table 7: SSD Training Parameters

Next, for the remaining figures that follow in this subsection, we visually show the latent features extracted via SSD for the adversarial samples and the original benign samples (for all models and all datasets), and find that they are very similar (refer to Figure 43 for CIFAR-10, Figure 44 for CIFAR-100, and Figure 45 for STL-10). This further supports our findings of low detection rates of our adversarial samples (as seen in the main text).

---

[4]https://github.com/inspire-group/SSD

|  | Anomalies | CIFAR-10 | CIFAR-100 | STL-10 |
|---|---|---|---|---|
| CC | Injected | $5139.84 \pm 810.542$ | $4920.48 \pm 896.119$ | $3889.28 \pm 616.61$ |
|  | Detected | $264.44 \pm 44.57$ | $297.65 \pm 52.89$ | $689.9 \pm 112.9$ |
| MICE | Injected | $4969.6 \pm 767.541$ | $5014.72 \pm 823.082$ | $3934.08 \pm 652.76$ |
|  | Detected | $1069.14 \pm 165.72$ | $664.22 \pm 110.656$ | $1024.38 \pm 168.88$ |
| NNM | Injected | $5017.6 \pm 792.99$ | $5044.32 \pm 711.296$ | $4021.12 \pm 649.94$ |
|  | Detected | $1621.51 \pm 259.8$ | $1897 \pm 270.715$ | $583.21 \pm 97.03$ |
| SCAN | Injected | $5031.04 \pm 784.24$ | $5034.24 \pm 667.093$ | $4125.44 \pm 609.97$ |
|  | Detected | $255.16 \pm 40.597$ | $970 \pm 131.402$ | $139.47 \pm 22.69$ |
| RUC | Injected | $4910.88 \pm 828.528$ | $4903.04 \pm 808.163$ | $3918.08 \pm 664.27$ |
|  | Detected | $45.27 \pm 8.68$ | $487.41 \pm 83.014$ | $209.51 \pm 36.58$ |
| SPICE | Injected | $4968 \pm 818.553$ | $4954.72 \pm 774.21$ | $3852.16 \pm 791.75$ |
|  | Detected | $253.02 \pm 43.279$ | $1481.32 \pm 228.984$ | $133.8 \pm 27.674$ |
| Averaged over 100 runs and 95 percentile cutoff | | | | |

Table 8: SSD Results

### B.2.1 CIFAR-10

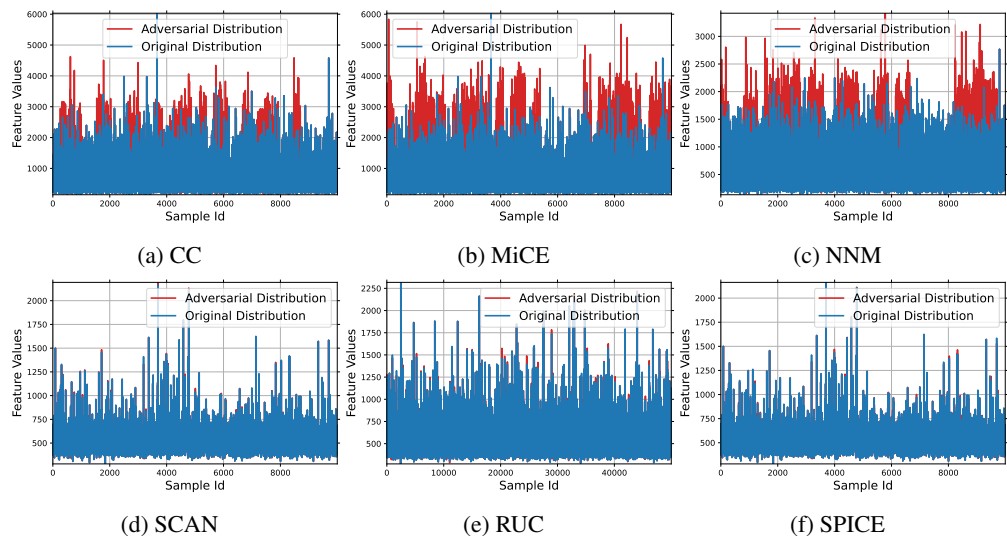

(a) CC  (b) MiCE  (c) NNM

(d) SCAN  (e) RUC  (f) SPICE

Figure 43: SSD Anomaly Detection (CIFAR-10)

### B.2.2 CIFAR-100

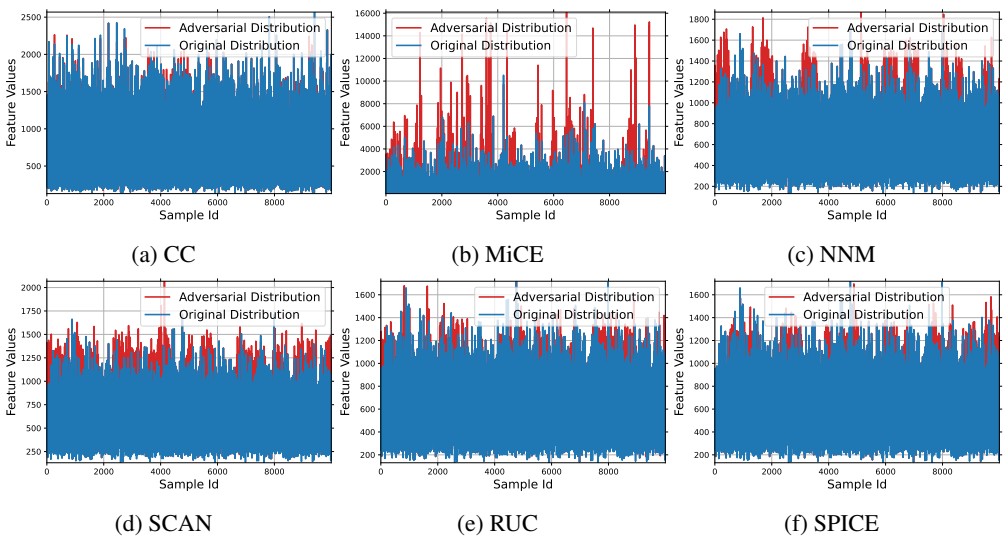

(a) CC

(b) MiCE

(c) NNM

(d) SCAN

(e) RUC

(f) SPICE

Figure 44: SSD Anomaly Detection (CIFAR-100)

### B.2.3 STL-10

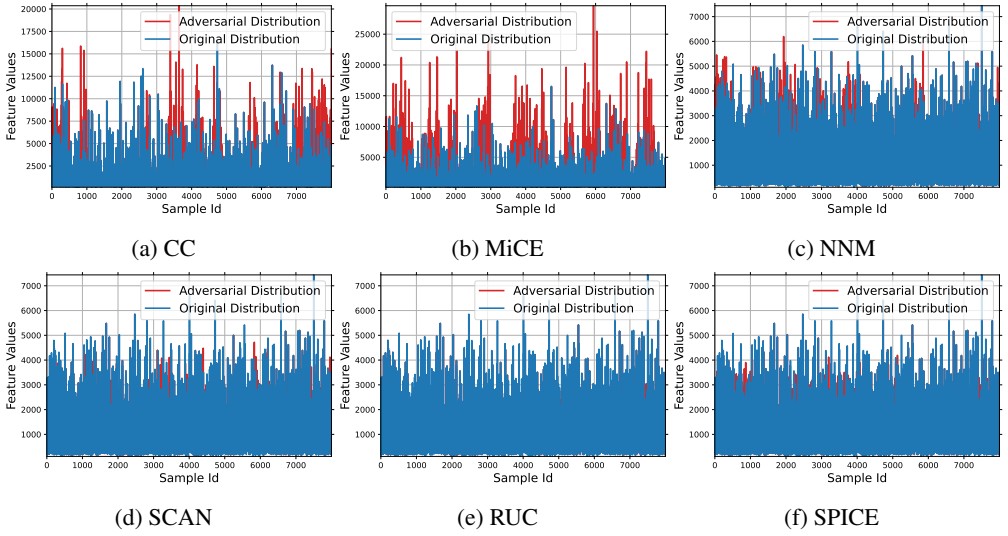

(a) CC

(b) MiCE

(c) NNM

(d) SCAN

(e) RUC

(f) SPICE

Figure 45: SSD Anomaly Detection (STL-10)

## C   Additional Results for `Face++` and Implementation Details

As mentioned in the main text, we attack the `Face++` API using a surrogate model. For these experiments we consider the Extended Yale Face Database B [17], and train CC on this dataset as the surrogate model. We use our proposed GAN framework to train the adversarial noise generator and attack the CC surrogate model. We then use this trained generator to generate noise for images that we provide to the `Face++` API. The training hyperparameters for CC are listed in Table 9.

|  | learning rate | epochs | seed | batch size |
|---|---|---|---|---|
| CC | 0.0003 | 1000 | 42 | 128 |

Table 9: Implementation parameters for Training CC for `Face++`

We also provide the intermediate pre-attack and post-attack values obtained using our GAN attack against CC. These are shown in Table 10.

|  | NMI | ARI | ACC |
|---|---|---|---|
| Before attack | 0.30 | 0.14 | 0.18 |
| After attack | 0.14 | 0.03 | 0.12 |

Table 10: Performance Values

For our attack, we employ the following API endpoints from `Face++`. To generate the API keys we use the `user console`[5] which allows us to set up a free account (albeit with rate limitations).

Then to create an album (that is, the dataset for clustering) we use the `CreateAlbum`[6] endpoint. This returns a `facealbum_token` which is used as an identifier for that album. Then we use the `AddImage`[7] endpoint to iteratively upload the images to the album. After uploading all the faces, we invoke the clustering process on the assigned album by calling the `GroupFace`[8] API. Finally, we call `GetAlbumDetail`[9] to get the assigned groups (that is, clusters) for all the faces.

### C.1 Sample Adversarial Images

In this subsection, we provide some sample adversarial images (and ground truth images) for the `Face++` attack. Figure 46 shows all the 28 ground truth faces that constitute the Extended Yale Face B dataset. Then, Figure 47 shows the adversarial samples used for the attack, and their pre-attack/post-attack labels.

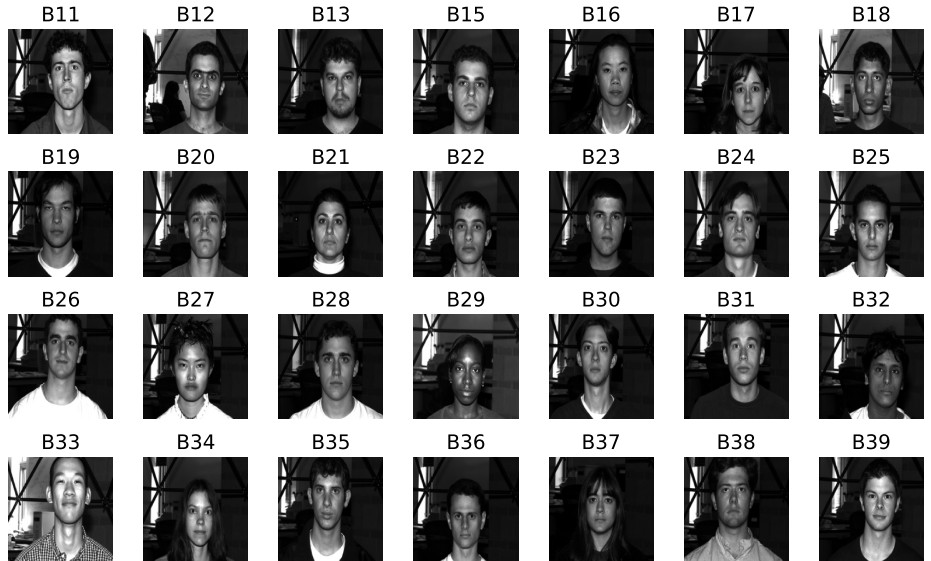

Figure 46: Ground truth faces for attack on `Face++`.

---

[5]https://console.faceplusplus.com/login

[6]https://console.faceplusplus.com/documents/53546559

[7]https://console.faceplusplus.com/documents/53546547

[8]https://console.faceplusplus.com/documents/53546579

[9]https://console.faceplusplus.com/documents/53546571

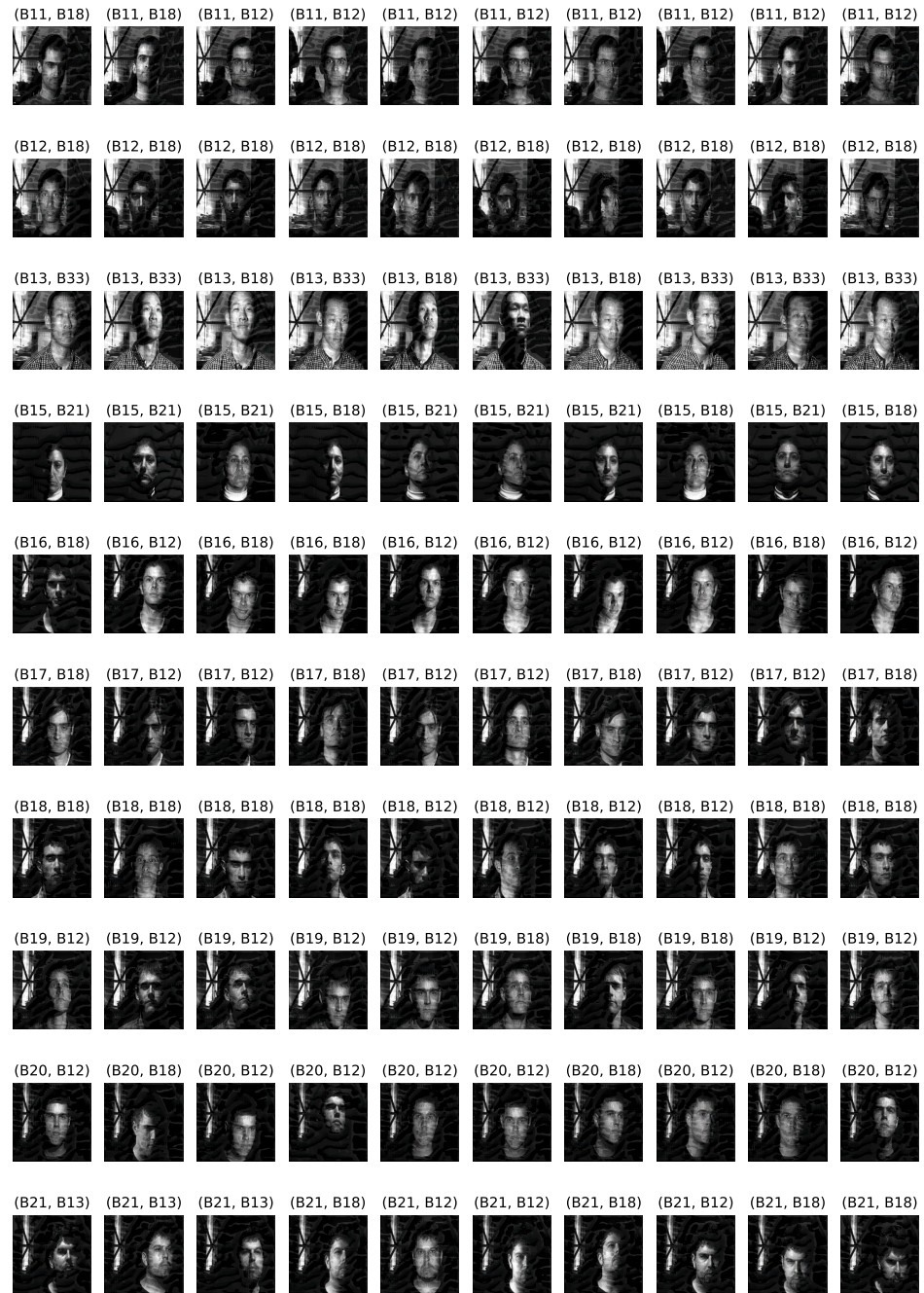

Figure 47: Adversarial samples for `Face++` (note that image labels here are a tuple indicating the pre-attack cluster on the left, and the post-attack cluster label on the right).

## D   Code, and System Details

**System Details:** All of the experiments were run on a 10 core Intel(R) Xeon(R) Silver 4114 CPU @ 2.20GHz with 32 GB of graphics memory (VRAM) on Tesla V100-PCIe GPU.

**Code:** The entire codebase is provided as a separate `zip` file in the supplementary material. We provide a `README` at the top-level directory which details the steps for reproducing our results using our code. We also provide all the models that we have trained (either via training scripts or as

pre-trained models). For original deep clustering pre-trained models from existing works we either provide training scripts or defer the reader to the original model repositories.

# E  Additional Experiments Comparing With SPSA (Supervised Blackbox Attack)

We undertake additional experiments to compare our GAN attack with the SPSA attack [18], which is a score-based blackbox attack against supervised models. We compare our attack with SPSA (original paper [18] loss) and SPSA (modified loss where predicted labels are used as ground truth) for all datasets and the SPICE (SOTA #1) and RUC (SOTA #2) models. We set common attack parameters according to the CIFAR-10 challenge parameters[10]. Note that while our perturbation norm threshold ($\epsilon$) is defined using the $l^2$ norm, the SPSA attack optimization problem defines their norm threshold using the $l^\infty$ norm. Thus, we use the same $\epsilon$ for our attack as in the rest of the paper, and for SPSA we use the values described in the CIFAR-10 challenge.

We use the SPSA attack implementation provided in the `advertorch` package[11]. We limit both our attack and the SPSA attack to $400$ blackbox queries to the deep clustering model. Note that for fair comparison query counts can be calculated for our attack as: $\frac{(\#\ \text{data samples}) * (\#\ \text{iterations})}{(\#\ \text{batch size})}$ and for SPSA as: $\frac{(\#\ \text{data samples}) * (\#\ \text{iterations}) * 2}{(\#\ \text{batch size})}$. There is an additional factor of 2 for SPSA as it makes 2 queries per iteration (refer to line 5 ("Calculate $g_i$") of Algorithm 1 in [18]).

Results for CIFAR-10 are provided in Table 11, for CIFAR-100 in Table 12, and for STL-10 in Table 13. As can be seen in the results, our GAN attack performs much better compared to the SPSA attack by a wide margin for all datasets and both models. This is possibly due to the difference in loss functions, as our GAN attack uses a stronger loss formulation via $\mathcal{L}_{\text{attack}}$. Furthermore, for each dataset this trend holds, and SPSA is especially unsuccessful for STL-10. To reiterate, SPSA with the original loss performs the worst, and our attack always outperforms SPSA even with modified loss.

Table 11: Results for CIFAR-10

| Model | RUC | | | SPICE | | |
|---|---|---|---|---|---|---|
| | NMI | ARI | ACC | NMI | ARI | ACC |
| Pre-attack | 0.81 | 0.79 | 0.89 | 0.85 | 0.84 | 0.92 |
| SPSA (original loss with ground truth labels) | 0.68 | 0.64 | 0.80 | 0.86 | 0.84 | 0.92 |
| SPSA (modified loss with predicted labels) | 0.70 | 0.67 | 0.83 | 0.64 | 0.60 | 0.78 |
| GAN Attack (ours) | **0.49** | **0.36** | **0.65** | **0.24** | **0.12** | **0.40** |

Table 12: Results for CIFAR-100

| Model | RUC | | | SPICE | | |
|---|---|---|---|---|---|---|
| | NMI | ARI | ACC | NMI | ARI | ACC |
| Pre-attack | 0.55 | 0.39 | 0.53 | 0.57 | 0.39 | 0.54 |
| SPSA (original loss with ground truth labels) | 0.39 | 0.24 | 0.41 | 0.57 | 0.41 | 0.54 |
| SPSA (modified loss with predicted labels) | 0.37 | 0.25 | 0.43 | 0.39 | 0.26 | 0.42 |
| GAN Attack (ours) | **0.24** | **0.07** | **0.27** | **0.14** | **0.02** | **0.17** |

Table 13: Results for STL-10

| Model | RUC | | | SPICE | | |
|---|---|---|---|---|---|---|
| | NMI | ARI | ACC | NMI | ARI | ACC |
| Pre-attack | 0.78 | 0.74 | 0.87 | 0.87 | 0.87 | 0.94 |
| SPSA (original loss with ground truth labels) | 0.77 | 0.74 | 0.86 | 0.80 | 0.79 | 0.89 |
| SPSA (modified loss with predicted labels) | 0.76 | 0.72 | 0.85 | 0.81 | 0.79 | 0.90 |
| GAN Attack (ours) | **0.22** | **0.05** | **0.29** | **0.32** | **0.11** | **0.41** |

---

[10]`https://github.com/MadryLab/cifar10_challenge`
[11]`https://github.com/BorealisAI/advertorch`

# F   Discussion on Inapplicaility of Supervised Attacks/Defenses

## F.1   Inapplicability of Supervised Blackbox Attacks

• **Absence of ground truth labels:** NES [19] and SPSA [18] cannot directly be used to attack deep clustering models based on their original formulation since ground truth labels are not available to the adversary and assuming so invalidates our threat model. In the SPSA paper, this can be seen in the attack optimization (Section 4.1) as $y_0$ is the ground truth label. Moreover, deep clustering output labels may not map to actual ground truth labels and hence, the attack will be unsuccessful.

• **Supervised attack loss with predicted labels and our proposed loss are different:** Even if we use a modified formulation for the supervised attack loss by replacing $y_0$ with the label predicted by the model, the SPSA margin and our loss $\mathcal{L}_{\text{attack}}$ are very different. The margin loss aims to reduce performance by ensuring that $y_0$ is not predicted for the adversarial sample, i.e. $j \neq y_0$ is instead predicted. Whereas our $\mathcal{L}_{\text{attack}}$ loss used for the GAN attack aims to maximize the probability that the *least likely* cluster label is assigned to a generated adversarial sample. This can be understood from the definition of the loss function, as it aims to maximize the *distance* between the *fixed* pre-attack cluster probabilities and the cluster probabilities of the adversarial sample. This is clearly a more powerful attack. We also believe that deep clustering models tend to be very over-/under-confident of certain classes when assigning cluster probabilities. This behavior is different from supervised models where errors are usually equitably distributed across classes and this possibly originates from the clustering task itself, i.e., lack of labels being used for training. Thus, minimizing the SPSA loss might not reduce the performance of the deep clustering model at all. Consider an example with 4 points and pre-attack predicted labels and ground truth labels are both [1,2,3,4]. For SPSA, the attack will be considered successful for an output [4,3,2,1] since all labels are different post-attack and classification accuracy is 0. However, the NMI before and after the attack is still 1.0, which is the best possible clustering achievable. Combining the ideas we discussed above, for our attack the output will more likely be [1,1,1,1] (or equivalent) where the NMI is 0, indicating *clustering breakdown*.

• **Question of choice of optimizer:** NES and SPSA attacks just utilize decades old finite gradient estimate techniques for carrying out their attack. At the end of the day, both our GAN and SPSA/NES approaches are derivative-free optimizers, so the choice of optimizer is largely irrelevant if our loss is used. This notion is also touched upon in the SPSA paper (paragraph above Section 4.2). Also, we expect GANs to work better as they are powerful deep learning models.

## F.2   Inapplicability of Supervised Learning Defenses:

The same problems exist for supervised defenses in the clustering context as defenses need to be truly unsupervised. Also, outputs of deep clustering models do not 1-to-1 map with ground truth labels, which is also an issue. We discuss popular defenses here from a recent adversarial attack/defense survey paper [20] (specifically Section 3.2 in [20]): 1) Robust training for deep clustering has been considered in our paper and shown to not be satisfactory (RUC), 2) Input transformation leads to shattered/vanishing gradients which cause further issues, 3) Randomization and 4) Model Ensembles do not work for deep clustering since labels do not correspond to ground-truth or some fixed label set (i.e., a label for one deep clustering model could be a very different representative cluster for another deep clustering model) so it is not trivial to combine models and doing so would be out of the scope of this work. 5) Certified defenses do not exist for deep clustering models.

# G   Additional Defense Experiments Using Earlier Deep Clustering Model and PCA

We undertake the following experiments: 1) The first experiment is based on utilizing an older version of a deep clustering method and attacking it using our GAN attack. The low-dimensional representations of benign and adversarial samples are visualized using t-SNE [21] and then analyzed using PCA as an anomaly detection approach. 2) The second experiment uses a SOTA deep clustering model and decomposes its benign and adversarial samples using the same older deep clustering method to low-dimensional representations. We perform the same t-SNE visualizations and PCA anomaly detection analysis on the benign and adversarial samples' embeddings.

**Design Choices For Experiments:**

• To use SSD for these experiments we would need to retrain the SSD model on the entire dataset of embeddings and there is no guarantee the SSD model would generalize well to these. Thus, instead of SSD, we use PCA for anomaly detection as it runs expediently and has a long history of being used for outlier/anomaly detection, especially on low-dimensional data [22].

• We undertook both experiments using the Deep Clustering Network (DCN) [23] as the low-dimensional embedding approach and the MNIST dataset. We utilized DCN since its architecture improved upon previous approaches by opting for an autoencoder as opposed to just an encoder, resulting in better clustering performance. Using MNIST with DCN is a suitable choice because 1) the original DCN implementation expects grayscale images, and 2) experimentally, DCN does not tend to work that well with more complex datasets such as CIFAR-10/CIFAR-100 (the authors also consider *only* MNIST in their paper). We use Contrastive Clustering (CC) as the SOTA deep clustering model to attack since it trains much quicker compared to other SOTA models.

|  | NMI | ARI | ACC |
|---|---|---|---|
| Pre-attack | 0.72 | 0.64 | 0.77 |
| Post-attack | 0.03 | 0.00 | 0.10 |

Table 14: DCN Performance Results

### G.1 Experiment on Attacking DCN:

• We train DCN on MNIST and attack it using the GAN attack in the paper (Table 14). Note that the dimension of the latent space in DCN is set to 10, and hence, we will effectively reduce high-dimensional image data samples to now have 10 features.

• We then use the low-dimensional representations obtained using the DCN network for both the benign and adversarial samples, and run t-SNE to visualize this low-dimensional space. These results are shown in Figure 48.

• As can be seen in Figure 48b and Figure 48c, that while t-SNE visualizations of the benign and adversarial samples look different visually, it is still hard to tell these apart unless the adversarial samples are known *a priori* (Figure 48a shows both superimposed). Figure 48d shows how different the predicted cluster labels are for the adversarial samples.

• We then run PCA on these embeddings (60000 benign and 60000 adversarial samples) jointly as an anomaly detection approach to see how many adversarial samples are detected overall. While this direction as a possible defense seems promising, a large majority of samples are still not detected with only 20% (12000 out of 60000) being detected.

• The reason for this is not surprising, as can be seen in Figure 48d that the predicted labels for the adversarial samples are mostly cluster label 8, indicating clustering breakdown.

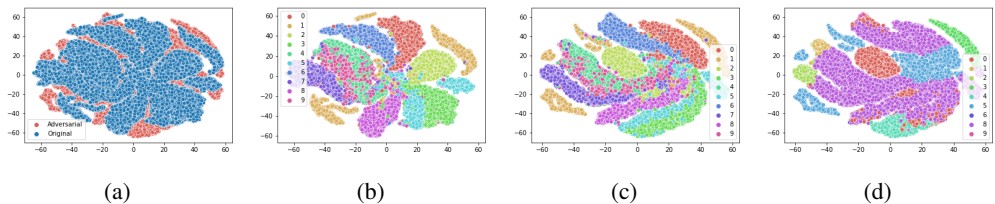

Figure 48: Visualizing DCN Benign and Adversarial Sample Embeddings: (a) Adversarial and Original Sample Embeddings, (b) Original Sample Embeddings with Ground Truth Labels, (c) Adversarial Sample Embeddings with Ground Truth Labels, (d) Adversarial Sample Embeddings with Predicted Labels

|  | NMI | ARI | ACC |
|---|---|---|---|
| Pre-attack | 0.72 | 0.64 | 0.77 |
| Post-attack | 0.28 | 0.09 | 0.30 |

Table 15: CC Performance Results

### G.2 Experiment on Attacking CC:

• We train CC on MNIST and conduct the GAN attack on this model to obtain adversarial samples (Table 15).

• We then obtain embeddings of both these benign and adversarial samples using the DCN model trained on MNIST.

• We carry out the same analysis by generating t-SNE visualizations for benign sample and adversarial sample embeddings. The results are shown in Figure 49 and a similar trend follows from the previous experiment. It is extremely hard to distinguish between adversarial and benign sample representations. This is observable in Figure 49b and Figure 49c. In fact, the adversarial samples here for CC are much closer to benign sample embeddings than the adversarial samples crafted for DCN specifically.

• Then, we run PCA on the combined benign and adversarial samples for anomaly detection. Here too, the same trend holds. Approximately, slightly less than 20% of adversarial samples are detected (11981 out of 60000). Thus, while the suggested approach has potential as a defense strategy, it is unable to detect a large number of adversarial samples.

• This is also as we expect, as there is clustering breakdown again for cluster label 8, which now has a large majority of samples.

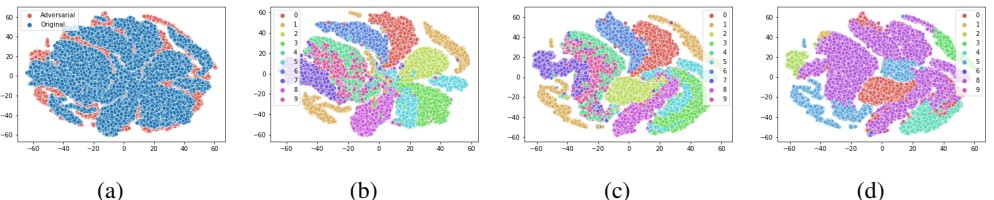

|  (a)  |  (b)  |  (c)  |  (d)  |

Figure 49: Visualizing CC Benign and Adversarial Sample Embeddings: (a) Adversarial and Original Sample Embeddings, (b) Original Sample Embeddings with Ground Truth Labels, (c) Adversarial Sample Embeddings with Ground Truth Labels, (d) Adversarial Sample Embeddings with Predicted Labels

## H   Real-World Practical Attack Scenarios

We discuss two possible real-world attack scenarios, where a malicious adversary can utilize adversarial attacks against deep clustering, under our threat model setting:

### H.1   Malware Analysis

A number of malware analysis tools (such as the popular Malheur [24], or [25, 26]) use cluster analysis in the backend. Assume an adversary exists that can query these systems and seeks to attack them by disrupting their clustering capability. Since clustering models are being utilized (and given that cluster probabilities are available), our GAN attack can be used to minimally perturb malware samples such that they change cluster membership and are instead clustered with benign samples. If the adversary releases this *perturbed* malware, it is unlikely that it will be clustered appropriately by these tools. Note that even if cluster probabilities are unknown or if the attacker cannot always query the system, a surrogate model can be trained for which adversarial samples can be generated (as we did for the `Face++` API) to carry out a transferability attack. Finally, there are other considerations for this attack– often malware clustering is not a binary clustering problem and many different clusters exist to represent different malware types. Thus, the attacker has even more freedom to disrupt these tools, and defending against such attacks would be a challenging problem.

### H.2   Human Activity Recognition (HAR) Using Wearables

Very recently, deep clustering has been successfully applied to the problem of Human Activity Recognition (HAR) where wearable sensors are used to record the user's data [27] and their current "activity" is predicted using the clustering model. While most research on HAR comprises of

supervised approaches, often it is impossible to collect and annotate data for this task, prompting the use of deep clustering. Moreover, the target audience/users for HAR are generally elderly people [28] as it can be used to prevent unprecedented health risks (such as falls, etc.). Given that deep clustering is being utilized for this task, an informed adversary can use this to cause direct harm to the users if they wish to do so. If they can gain control (i.e., query the system) of the target's wearable device (through social engineering, security flaws, etc.), they can generate adversarial samples for it using our attack. In this manner, even if an individual needs immediate help, the attacker can use adversarial samples (which will likely be misclustered) to make the cause (activity) seem benign. The contrary case is also problematic– the adversary can generate adversarial samples for multiple devices that lead to "false alarms" solely for anarchistic purposes. Defending against such an attack would also be significantly challenging.