# OpenReview forum: "On the Robustness of Deep Clustering Models: Adversarial Attacks and Defenses"
_NeurIPS.cc/2022/Conference — NeurIPS 2022 Accept_

### Official Review · Reviewer_88S9 · 2022-07-02

**Rating:** 6
**Confidence:** 4
**Soundness:** 3 good
**Presentation:** 4 excellent
**Contribution:** 3 good

**Summary:**

This paper proposes the first black-box test-time adversarial attack against deep clustering models. This work helps bridge the adversarial analysis from traditional clustering approaches (explored before) to deep learning-based clustering models, where only white-box supervised learning based attacks have been explored. The attack uses Generative Adversarial Networks (GAN) with just score-based query access to the deep clustering model, with a more clearly defined threat model. The authors evaluate this attack on SOTA deep clustering models, in a transferability study, and on the Face++ ML-as-a-Service (MLaaS) system. The authors also demonstrate that existing defense approaches fail to stop the attack.

**Questions:**

The paper is well-written and explores an interesting problem (the adversarial robustness of deep clustering techniques along with some mitigation strategies). Overall, I think this is a good submission, and am encouraged to see work on black-box attacks on real MLaaS systems.

One aspect that could be improved is in making further distinctions between the authors' attack and AdvGAN, as it seems that there are many technical similarities between AdvGAN and the GAN formulation in this work. Although it is adapted somewhat to the clustering problem, the overall formulation with an adversarial, hinge, and GAN loss is very similar to AdvGAN, limiting its technical novelty.

While generally well-written, there remain some things that could be further clarified in the paper. How is the surrogate to Face++ trained? Also, how are the epsilon bounds set for the table? It seems arbitrary given that the numbers in the Appendix are not consistent across datasets and it is unclear why they are different, so it seems some justification would help. Also in this case then, for the transferability studies, are the epsilons still unequal?

Do you have any insight into: 1) why there seems to be common specific patterns in the noise and 2) why most attacks are clustered in the same class?

Finally, I think more examples of specific real-world threat vectors could be included to better: 1) scope the societal impact of the work and 2) better motivate the need for robust deep clustering techniques.

Minor comments:
- Could clarify a bit what the specific threat model contribution is when brought up in the Intro (line 46)
- Intro: could cite some specific numbers on percentage of performance changes with the attack
- Could further distinguish between when a "score-based" black-box model is used vs. a "decision-based" black-box model


**Limitations:**

The authors have provided clear descriptions on the level of access needed to data, models, etc. to pull off the attack, and several mitigation approaches are mentioned and studied. The authors also discussed their hope that this work leads to successful development of defenses. Specific examples of real-world motivations and consequences of attacks on deep clustering algorithms could be further discussed.

**Strengths And Weaknesses:**

Strengths:
- Attacks a real-world MLaaS
- Evaluates against potential mitigation strategies
- Strong attack results

Weaknesses:
- Limited technical novelty
- Parts of the problem have been previously explored (e.g., traditional clustering, or white-box attacks on deep clustering), although the combination of factors is new
- Ambiguous epsilon bounds

---

> ### Author Response · Authors · 2022-08-02
> **Authors' Response to Reviewer 88S9**
>
> We thank the reviewer for their thorough reading of the paper and feedback. We have answered the questions below:
>
> - __Improvements over Vanilla GAN/AdvGAN:__ 1) We propose a novel loss function ($L_{attack}$) that aims to completely disrupt the deep clustering function by reducing the NMI/ACC/ARI of the models significantly. 2) The AdvGAN approach is designed for supervised learning, and the authors used distillation [1] for the blackbox attack which is invalidated by our threat model, further simplifying our approach. 3) Unlike AdvGAN, we implement adversarial image clipping [2] so values are within the appropriate range (0 to 255). This improves performance and generates more realistic adversarial images.
> - __Face++ Surrogate Model:__ We have provided training details for the CC [3] model that is used as the _surrogate_ for the Face++ attack in Appendix C. We train the CC deep clustering model on the Yale Face B dataset. Then, we use the GAN attack to generate attack samples for the CC model we trained on Yale Face B. The same set of benign and adversarial images (generated by GAN attack on CC) are provided to the Face++ API. We can then measure performance (NMI) pre-attack and post-attack to calculate the efficacy of our attack on Face++.
> - __Choice of $\epsilon$:__
>     - Our experiments to determine the best $\epsilon$ values were undertaken using grid search. However, the optimal perturbation found by the GAN might be much lower than $\epsilon$. Thus, to report more precise $\epsilon$ values we calculated the norm of the optimal perturbation and report that as $\epsilon$ in the paper. We will refer to this here as $\epsilon^*$. This is why the reported values seem somewhat arbitrary, and we apologize for not detailing this process in the original submission; we omitted it believing it is an implementation detail. In the final version we will include both and add a remark regarding this.
>     - Using the reported values of $\epsilon^*$ for the attack instead of $\epsilon$ generates the same adversarial samples, as $\epsilon^* \leq \epsilon$ and no extra clipping can occur. We confirmed this by re-running experiments before submitting. Note that for some models/datasets the GAN network converges much quicker, resulting in widely varying $\epsilon$ and $\epsilon^*$ values.
>     - The images used for the attack are the ones visually reported in Appendix A.2 so it can be seen that the noise threshold is low from a visual perception sense.
> - __Regarding $\epsilon$ values for the transferability table:__ While $\epsilon$ can be fixed, for each model/dataset, the GAN attack might end up finding different optimal noise with different norm, i.e. different $\epsilon^*$. Thus the adversarial perturbation norm values ($\epsilon^*$) are different for the transferability matrix. We do not think this is a problem as previous foundational work on transferability also had different perturbation norm for different models (Section 3.3 in [4]).
> - __Possible reasons for clustering breakdown:__ We feel deep clustering models tend to be very over-/under-confident of certain classes when assigning cluster probabilities. Our $L_{attack}$ loss for the GAN attack is powerful as it aims to maximize the probability that the _least likely_ cluster label is assigned to a generated adversarial sample by maximizing the _distance_ between the _fixed_ pre-attack cluster probabilities and the cluster probabilities of the adversarial sample. Considering the above, the clustering breakdown effect seen in the paper possibly occurs because a) our loss is driving the model to pick the worst cluster label possible for the sample, and b) for deep clustering models the least likely classes/clusters are not equi-probable, i.e., one or a few cluster labels are generally the least likely cluster labels for a majority of samples. For e.g. consider 4 data samples with ground truth labels [1,2,3,4]. The clustering model outputs [1,2,3,4] pre-attack, giving NMI=1. For our attack the output will more likely be of the type [1,1,1,1] where NMI=0.
> - __Addition of real-world threat scenarios:__ We have added real-world threat vectors for our attack in Appendix H. We provide attack scenarios pertaining to 1) Malware Analysis, where clustering models are heavily used, and 2) Human Activity Recognition. We have also added more details to our limitations (Section 7).
> - __Minor comments:__ We have addressed the minor comments suggested by the reviewer in the main text. Please see line 46 (footnote 2) regarding threat model contribution, lines 84-85 (footnote 4) regarding % performance changes, and line 145 (footnote 6) regarding distinction between score-based and decision-based attack.
>
> References:
> 1) G Hinton, et al. Distilling the knowledge in a neural network. 2015.
> 2) N Carlini, et al. Towards evaluating the robustness of neural networks. 2017.
> 3) Y Li, et al. Contrastive clustering. 2021.
> 4) N Papernot et al., Transferability in Machine Learning. 2016.

---

> > ### Comment · Reviewer_88S9 · 2022-08-05
> > **Response to Authors**
> >
> > Thank you for your detailed and well-prepared response to my concerns on differences between Vanilla GAN / Adv GAN, Face++ surrogate model details, epsilon values, clustering intuitions, and real-world threats, which clears up my concerns. I think a score of 6 remains fair, as a solid paper that provides an interesting study that applies to real-world black-box MLaaS systems with a novel but inspired by prior work technique.

---

> > > ### Author Response · Authors · 2022-08-06
> > > **Author Reply**
> > >
> > > Thank you for taking the time to go through the rebuttal/revision and for your constructive feedback, we appreciate it.

---

### Official Review · Reviewer_sgLu · 2022-07-10

**Rating:** 5
**Confidence:** 5
**Soundness:** 2 fair
**Presentation:** 2 fair
**Contribution:** 2 fair

**Summary:**

This paper presents a new black-box attack against deep clustering models. Specifically, the adversarial perturbation is computed by a generative model. It penetrates existing defense methods for clustering, and manifests transferrability which further make the proposed method practical. To further demonstrate the effectiveness of the proposed method, experiments are also conducted on a commercial API named Face++, apart from commonly used evaluation datasets in previous related works.

**Questions:**

My questions are written in weaknesses 4, 5, 6, and 8.

**Limitations:**

Limitations are not discussed in detail -- only several words in section 7.
Potential societal impact is discussed in the last paragraph. I agree that in order to be able to defend, we first have to know how to attack.

**Strengths And Weaknesses:**

# Strengths

1. The manuscript focuses on run-time adversarial attack on deep clustering model, which is a less-explored area.

2. The proposed attack method is simple and intuitive. Experiments demonstrate its effectiveness.

# Weaknesses

1. [minor issue: paper organization] The introductory part of the manuscript can be improved. Since the proposed attack and defense methods are all based on deep learning, it is no longer necessary to introduce the clustering task starting from traditional methods like k-means. Instead, it is suggested to briefly mention (1) how a deep learning-based clustering method works, (2) how does attack against such method work, and (3) the potential risk induced by the attack to practical applications. These will make it more worthwhile for this topic to draw attention from the community (line 40).

2. [minor issue: unclear figure] In figure 1, it is unclear which picture correspond to which model mentioned on line 79 (SPICE, RUC).

3. [minor issue: potentially misleading] On line 35, it is pointed out that “zero-th order optimization is intractable for deep models.” in foot note. However, zero-th order optimization is a common approach for black-box attack against supervised deep classification models. See survey paper “Benchmarking Adversarial Robustness on Image Classification, CVPR2020” for examples, including ZOO, NES, as well as SPSA. In this sense, the word “intractable” is ambiguous, and may possibly convey wrong information to the reader.

4. [insufficient citations] This paper focuses on attack and defense problem on a specific domain other than classification. In section 2, the author should cite and discuss the main-stream attack defense works on the classification problem, and discuss how and why these ideas cannot be adapted for clustering.

5. [misleading contents] In the field of classification, “black-box attack” usually refers to run-time attacks against a trained and frozen model. See “Benchmarking Adversarial Robustness on Image Classification, CVPR2020” for examples. The attacks happend during training-time are called Backdoor attack or Poisoning attack. So please avoid using “adversarial attacks against clustering” as paragraph title on line 90. To be less ambiguous, it should be “Training-time attacks against ...”.

6. [important: relationship with classification attack] Based on the description on line 116. A frozen trained clustering model is a classification model to assign cluster labels from softmax probabilities. In this way the existing black-box (run-time) attacks should be compatible in the setting of this paper. However, comparison and discussion are missing. The aforementioned CVPR2020 survey paper can be used as a referece regarding this. Existing (black-box, run-time) classification attack methods can be used to attack the cluster assignment, e.g. based on the softmax scores. To be clear, score-based attacks like NES and SPSA in the mentioned CVPR2020 paper can do this. In this sense, the claim on line 138 is wrong. Score-based attack like NES and SPSA can optimize M (softmax scores) using estimated gradients even if the optimization target is completely indifferentiable.

7. [minor issue: unconvincing claim] On line 210, the ALRDC approach makes the latent space robust to perturbations. A robust latent space is already very important for adversarial robustness, and it not clear why this restricts its use. That said, evidence provided on line 212 indeed makes comparison with ALRDC unnecessary.

8. [confusing figure] Figure 2 shows an irregular pattern after attack. According to equation on line 148, the attack is untargeted, which means the objective does not care which would be the result class for misclassification. In this sense, shouldn't the confusion matrix be relative random instead of being concentrated on the “cat” or “horse” class?

9. [misleading title] The title mentions “defense”. But the paper actually only involves attack and evaluated this attack against existing methods for defense. The paper does not propose any new possibility for defense.

---

> ### Author Response · Authors · 2022-08-02
> **Authors' Response to Reviewer sgLu**
>
> We thank the reviewer for their insightful feedback. We address the concerns in our response, and kindly request the reviewer to reconsider their score.
>
> __Inapplicability of supervised blackbox attacks:__
> - __Absence of ground truth labels:__ NES [1] and SPSA [2] cannot directly be used to attack deep clustering models since ground truth labels are not available to the adversary and assuming so invalidates our threat model. This is seen in the SPSA paper attack optimization [2] (Section 4.1) where $y_0$ is the ground truth label and in the suggested CVPR 2020 paper [3] (Section 2.1) where $y$ is the ground truth label. Moreover, deep clustering output labels may not map to actual ground truth labels and hence, the attack will be unsuccessful. In new experiments we did for all datasets (models: SPICE, RUC) using this setting, the SPSA attack was mostly unsuccessful and at times increased NMI (Appendix E, Table 11-13).
> - __Supervised attack loss with predicted labels and our proposed loss are different:__
>     - Even if we use a modified formulation for the supervised attack loss by replacing $y_0$ with the label predicted by the model, the SPSA margin loss and our loss $L_{attack}$ are very different. The margin loss aims to reduce performance by ensuring that $y_0$ is not predicted for the adversarial sample, i.e. $j\neq y_0$ is predicted. Whereas our $L_{attack}$ loss aims to maximize the probability that the _least likely_ cluster label is assigned to a generated adversarial sample. I.e., it aims to maximize the _distance_ between the _fixed_ pre-attack cluster probabilities and the cluster probabilities of the adversarial sample. This leads to a more powerful attack.
>     - We believe that deep clustering models tend to be very over-/under-confident of certain classes when assigning cluster probabilities. This possibly originates from the lack of labels during training. Hence, the least likely classes/clusters are not equi-probable. Further, minimizing the SPSA loss might not reduce performance of the deep clustering model at all.
>     - Consider an example with 4 points. The pre-attack predicted labels and ground truth labels are both [1,2,3,4]. For SPSA, the attack will be considered successful for an output [4,3,2,1] since all labels are different post-attack and classification accuracy is 0. However, the NMI before and after the attack is still 1.0, which is the best possible clustering achievable. For our attack the output will more likely be of the type [1,1,1,1] where NMI=0, indicating _clustering breakdown_.
>     - This failure of this supervised loss is shown in additional experiments (Appendix E, Table 11-13). For the same number of queries, we find that SPSA cannot reduce the clustering performance as well as our attack.
>
> __Concerns:__
> 1. Thank you for the suggestions. We added two real-world attack scenarios to the paper (Appendix H) and improved limitations (Section 7).
> 2. We corrected the Figure 1 caption.
> 3. To avoid ambiguity, we re-worded line 35.
> 4. We added a subsection "Attacks/Defenses in Supervised Learning" to Section 2, cited the relevant works (NES, SPSA, CVPR paper [3], etc.). We provide more details in Appendix F. As mentioned above (and Appendix E,F), classification attacks/defenses cannot be directly applied to deep clustering with good results.
> 5. We changed it to "Training-time Adversarial Attacks Against Clustering" to remove ambiguity.
> 6. Thank you for pointing out the oversight on line 138-- we have reworded it (line 153 in revised paper) so that it is correct. NMI/ACC/ARI require ground truth labels and are only used for evaluation so we cannot use them as the attack optimization target (even if they are non-differentiable). In the response above, we have detailed why supervised attacks still cannot be used successfully.
> 7. As mentioned in the main text not all deep clustering models have latent spaces (earlier versions did). As we state in line 210, ALRDC can make the latent space robust to perturbations. However, the SOTA deep clustering models considered in this paper do not possess a latent space.
> 8. We have hypothesized the reason for this trend in our response above. It seems that for SPICE/CC the model assigns a sample low probability w.r.t "horse"/"cat" unless it actually belongs to that cluster. Thus, these are usually the least likely cluster label (for SPICE/CC), and our attack moves samples to these labels, leading to very low clustering performance (NMI).
> 9. We believe the word "defenses" in the title does not imply that we propose a new defense and we explicitly state that we use existing defenses throughout.
>
> References
> 1) A Ilyas, et al. Black-box adversarial attacks with limited queries and information. 2018.
> 2) J Uesato, et al. Adversarial risk and the dangers of evaluating against weak attacks. 2018.
> 3) Y Dong, et al. Benchmarking adversarial robustness on image classification. 2020.
> 4) P Huang, et al. Deep embedding network for clustering. 2014.

---

> > ### Comment · Reviewer_sgLu · 2022-08-06
> > **Response to Authors**
> >
> > Thanks the authors for the detailed clarifications.
> >
> > The response (including appendix E) has clearly addressed my concerns, and made the difference between classification attack and the proposed method clear. I it highly encouraged to incorporate the justifications in the manuscript, especially the '[1,2,3,4] v.s. [4,4,4,4]' example. This really helps readers to understand the difference.
> >
> > Based on these, my major concerns are well-resolved. And I'm raising the score to 5.

---

> > > ### Author Response · Authors · 2022-08-07
> > > **Author Reply**
> > >
> > > Thank you for taking the time to go through the new results and response, and for updating the score. We will be sure to include the justifications and example in the final version.

---

### Official Review · Reviewer_hxny · 2022-07-11

**Rating:** 7
**Confidence:** 4
**Soundness:** 3 good
**Presentation:** 3 good
**Contribution:** 4 excellent

**Summary:**

This paper proposes an effective attack method against deep clustering models. Specifically, the threat model and attack objective are newly defined, reducing the attack to an optimization problem. Then, by combining a common GAN training loss with the attack objective, the authors design a GAN-based blackbox attack so that the resulting GAN generates adversarial perturbations for a given input image.

The power of the attack–effectiveness, query complexity, and transferability–is demonstrated from thorough experiments. Also, the authors tested the performance of their attack against two standard defense methods (robust deep clustering and deep learning based anomaly detection), showing successful attack results against them. Finally, they tested a blackbox attack against an existing face clustering service. Since the actual cluster membership is unknown in this case, the authors trained an open-source surrogate model and attacked via the transferability of the attack.


**Questions:**

1. While the paper demonstrated impressive performance against state-of-the-art deep clustering models, it would be interesting to attack some earlier deep clustering models. Since earlier models decompose the high-dimensional data to a low-dimensional representation, we can observe what happens in the low-dimensional space. (Those small perturbations in high dimensional may result in considerable differences in the low dimensional data.)
2. A follow-up suggestion from Question 1 is to perform anomaly detection over the low-dimensional representations of adversarial samples. To explain, prepare an earlier version of the deep clustering model as an embedding method. (Even though earlier versions have low performance in clustering tasks, it is okay because we will use it only to create low-dimensional representations) Then, create adversarial examples against a state-of-the-art model and embed those adversarial examples in low-dimensional space. Using any anomaly detection method, check whether the embedded representations are outliers or not.


**Limitations:**

The authors discussed limitations and potential negative societal impact in the paper appropriately.

**Strengths And Weaknesses:**

Originality: To the best of my knowledge, the paper contains novel ideas.

[[Strength]]
1. To the best of my knowledge, this is the first work about adversarial attacks against the deep clustering model.

Quality:

[[Strength]]
1. The authors tested the attack against numerous clustering models and performance metrics. To explain, for the attack performance test, the authors prepare six different state-of-the-art models and three different clustering performance measures.
2. Moreover, the authors also explore various aspects of the attack–query complexity and transferability–to better evaluate the attack’s value.
3. The power of the attack is evaluated carefully against existing defense strategies.

[[Weakness]]
1. Visualization of the PCA patterns between adversarial examples and original samples may not be an effective way to analyze the reason.  A better visualization should also show the difference in the PCA components between detected adversarial examples, undetected adversarial examples, and benign original samples. (The last two sets of samples should show similar patterns, while the first set should show visually different patterns.)

Clarity: The paper writing is clear, and there is no issue with the clarity of the paper.

Significance:

[[Strength]]
1. The power of attack looks to be very impressive. The attack significantly reduces the clustering performance for all 18 combinations (six different models, three different metrics).
2. The proposed attack seems to be effective against existing defenses. The experimental results may imply the need for a new defense method to ensure the robustness of deep clustering algorithms.
3. The authors experiment with the attack transferability and test it against an existing face clustering service (via an open-source surrogate model). This result shows an urgent threat of the attack in a practical setting.

---

> ### Author Response · Authors · 2022-08-02
> **Authors' Response to Reviewer hxny**
>
> We thank the reviewer for their insightful comments and suggestions, and hope to improve upon the weaknesses mentioned by doing some of the suggested additional experiments (Appendix G).
>
> __Additional Experiments:__
>
> We thank the reviewer for the excellent suggestion of using earlier deep clustering methods as an embedding approach. Based on this, we undertake the following experiments: 1) The first experiment (Appendix G.1) is based on utilizing an older version of a deep clustering method and attacking it using our GAN attack. The low-dimensional representations of benign and adversarial samples are visualized using t-SNE [2] and then analyzed using PCA as an anomaly detection approach. 2) The second experiment (Appendix G.2) uses a SOTA deep clustering model and decomposes its benign and adversarial samples using the same older deep clustering method to low-dimensional representations. We perform the same t-SNE visualizations and PCA anomaly detection analysis on the benign and adversarial samples' embeddings.
>
> __Design Choices For Experiments:__
> - Due to time constraints we cannot use SSD (the deep-learning based anomaly detection approach) for experiments since this model needs to be trained on the source/ground-truth data (i.e., all of CIFAR-10, or STL-10). To do this we would have to retrain the SSD model on the entire dataset of embeddings and there is no guarantee the SSD model would generalize well to these. Thus, instead of SSD, we use PCA for anomaly detection as it runs expediently and has a long history of being used for outlier/anomaly detection, especially on low-dimensional data [3].
> - We undertook both experiments using the Deep Clustering Network (DCN) [1] as the low-dimensional embedding approach and the MNIST dataset. The size of the latent space for DCN is set to 10. We utilized DCN since its architecture improved upon previous approaches by opting for an autoencoder as opposed to just an encoder, resulting in better clustering performance. Using MNIST with DCN is a suitable choice because 1) the original DCN implementation expects grayscale images, and 2) experimentally, DCN does not tend to work that well with more complex datasets such as CIFAR-10/CIFAR-100 (the authors also consider _only_ MNIST in their paper). We use Contrastive Clustering (CC) as the SOTA deep clustering model to attack since it trains much quicker compared to other SOTA models.
>
> __Experiment on Attacking DCN:__
>
> - We train DCN on MNIST and attack it using the GAN attack in the paper.
> - We then use the low-dimensional representations obtained using the DCN network for both the benign and adversarial samples, and run t-SNE to visualize this low-dimensional space. These results are shown in Figure 48 (Appendix G.1).
> - As can be seen in Figures 48(b),(c), while t-SNE visualizations of the benign and adversarial samples look different visually, it is still hard to tell these apart unless the adversarial samples are known _a priori_ (Figure 48(a) shows both superimposed).
> - We then run PCA on these embeddings (60000 benign and 60000 adversarial samples) jointly as an anomaly detection approach to see how many adversarial samples are detected overall (Appendix G.1). While this direction as a possible defense seems promising, a large majority of samples are still not detected with only 20\% (12000 out of 60000) being detected.
>
> __Experiment on Attacking CC:__
>
> - We train CC on MNIST and conduct the GAN attack on this model to obtain adversarial samples.
> - We then obtain embeddings of both these benign and adversarial samples using the DCN model trained on MNIST.
> - We carry out the same analysis by generating t-SNE visualizations for benign sample and adversarial sample embeddings. The results are shown in Figure 49 (Appendix G.2) and a similar trend follows from the previous experiment. It is hard to distinguish between adversarial and benign sample representations.
> - Then, we run PCA on the combined benign and adversarial samples for anomaly detection (Appendix G.2). Here too, the same trend holds. Approximately, slightly less than 20\% of adversarial samples are detected (11981 out of 60000). Thus, while the suggested approach has potential as a defense strategy, it is unable to detect a large number of adversarial samples.
> - Our future aim is to train a SOTA anomaly detection approach on the low-dimensional embeddings and undertake the same experiments on the main datasets considered in the paper.
>
> __References:__
> 1) B Yang, et al. Towards k-means-friendly spaces: Simultaneous deep learning and clustering. 2017.
> 2) L Van der Maaten et al. Visualizing data using t-sne. 2008.
> 3) C Aggarwal. An introduction to outlier analysis. 2017.

---

> > ### Comment · Reviewer_hxny · 2022-08-06
> > **Response to the authors**
> >
> > Thanks for the additional experiments on the low-dimensional embedding space. The result seems interesting, and it satisfied my curiosity about what is happening in the low-dimensional representation. Considering the high impact on one sub-area of AI (area: adversarial ML, and sub-area: adversarial attack on clustering algorithms), I believe the evaluation of 7 is fair for this paper.

---

> > > ### Author Response · Authors · 2022-08-06
> > > **Author Reply**
> > >
> > > Thank you for going through the additional experimental results and for your constructive feedback. We appreciate your time.

---

### Official Review · Reviewer_c5rd · 2022-07-26

**Rating:** 5
**Confidence:** 4
**Soundness:** 2 fair
**Presentation:** 3 good
**Contribution:** 2 fair

**Summary:**

This paper investigates the blackbox adversarial attacks against deep clustering models. It first proposes a blackbox adversarial attack using vanilla GAN. Then it evaluates the attacks against a number of SOTA deep clustering models and real-world images. It reveals that the proposed attack can significantly reduces the performance of these models, has minimal query complexity. Moreover, to further examine the robustness, it utilizes two unsupervised defense approaches, and observes unsuccessful defenses. And it also shows a successful attack on production-level API Face++.


**Questions:**

Besides the major questions in the "Weaknesses" part, there could be more deserving the authors' attentions:

3. The evidence of PCA might not be convincing. We assumes you perform PCA on the total datasets, including all classes. As a complicated clustering/classification task (not easily separable w.r.t. a few principle components), it is not supervising that all data mixed in PCA plot together with the adversarial ones (considering they are close to the original).

4. In Fig.6, why the "best" plot also has an errorbar. Did it show multiple "best"? How did you define the "best"? Maybe we missed something.

5. We failed to see the necessity or the supportiveness of the application of anomaly detection as a defense approach. Considering the adversarial attack samples are "close" to the original, it may be foreseen that they fail to be observed as anomalies.

6. The query complexity of the attack (Tbl.2) may not be a fair comparison. These attack mechanisms are different and the attacks are imposed at different stages of learning process. Not sure whether this comparison could tell something.


**Limitations:**

We do not see any potential negative societal impact of this paper.

**Strengths And Weaknesses:**

Strengths:

This paper, as the authors claimed, proposed the first blackbox adversarial attack against deep clustering models. And it performed multiple experiments to show its success on various SOTA models, real-world datasets, query complexity, transferability, resistance to defenses, and production-level API Face++.

Weaknesses:

1. The blackbox adversarial attacks seem a straightforward application of vanilla GAN to this problem. Although the authors mentioned the challenges in building GAN-based blackbox attacks in Sec.3, we did not see any novelty or smart operations in Sec.4.1 to build up blackbox attacks. And the most challenging part of the direct formulation of attack objective has been bypassed by practically choosing the indirect, and hence the ratio of successful attacks among total raises concerns.

2. The norm of adversarial noise, epsilon, may not be an appropriately interpretable one in experiments (Fig.3), since its reasonable threshold can vary significantly for different datasets. Obviously a large enough epsilon can always lead to a "successful" attack, while it may not be a meaningful one (it should be reasonably small such that this attack does not completely overwrite datasets). One has to formulate an index that quantifies the intensity of attacking noises over data.

---

> ### Author Response · Authors · 2022-08-02
> **Authors' Response to Reviewer c5rd**
>
> We thank the reviewer for their insightful comments and questions.
>
> 1. __Novelty of Attack Approach:__
>     - __Improvements from Vanilla GAN/AdvGAN:__ 1) We propose a novel loss function $\mathcal{L}_{attack}$ that ensures the NMI/ACC/ARI of deep clustering models decreases significantly. 2) The original AdvGAN approach was designed for supervised learning; the authors used distillation [1] for the blackbox attack which is invalidated by our threat model, making our approach simpler. 3) Unlike AdvGAN, we implement adversarial image clipping [2] so that values are within the appropriate range (i.e. 0 to 255). This improves performance and generates more realistic adversarial images. 4) Despite its simplicity, our attack significantly reduces the performance of deep clustering models, showcasing its disruptive capability.
>     - __The power of our loss function:__ We feel deep clustering models tend to be very over-/under-confident of certain classes when assigning cluster probabilities. Our $\mathcal{L}_{attack}$ loss for the GAN attack is powerful; it aims to maximize the probability that the _least likely_ label is assigned by maximizing the _distance_ between the _fixed_ pre-attack cluster probabilities and the adversarial sample's cluster probabilities. Thus, the clustering breakdown seen in the paper possibly occurs because a) our loss is driving the model to pick the worst possible cluster label for a sample, and b) for deep clustering models the least likely clusters are not equi-probable across samples, i.e., one or a few cluster labels are the least likely cluster labels for most samples. E.g. consider 4 data samples with ground truth labels [1,2,3,4]. The clustering model outputs [1,2,3,4] pre-attack, giving NMI = 1.0. Based on the described ideas, our attack would likely label all samples in the same cluster (e.g. [4,4,4,4]) making NMI = 0.
> 2. __Interpreting Norm of Adversarial Noise:__ The reviewer's recommendation is a great insight; one can define the _intensity_ of the noise $\epsilon$ added as $\frac{\epsilon}{\epsilon_{\max}}$ to make it more interpretable. Here $\epsilon_{\max}$ is the maximum possible noise added after which the image starts deteriorating, revealing it is an adversarial sample. We will add this as a note in the final version and could update figures as well with this index.
> 3. __Improving PCA:__
>     - We agree with the reviewer's suggestion. Our reason for undertaking the PCA analysis in the paper was to reinforce the results obtained via SSD (anomaly detection) and how it fails as a defense. However, based on the reviewers' suggestions, we improve the PCA analysis through a simple experiment using an older deep clustering model DCN [3] as a low-dimensional embedding approach. We obtain embeddings for both benign and adversarial samples for a SOTA model via DCN. We a) visualize the embeddings for benign and adversarial samples using t-SNE [4], and b) perform PCA based anomaly detection on the benign and adversarial sample embeddings to see if adversarial samples can be detected.
>     - __Implementation:__ We conduct this experiment on the CC [5] deep clustering model and generate adversarial samples for it. After, we use DCN to obtain the low-dimensional embeddings for benign and adversarial samples.
>     - __Results:__ a) The t-SNE visualizations show that while adversarial and benign samples look different visually (Figure 49(b),(c) in Appendix G.2) it is hard to tell these apart unless adversarial samples are known _a priori_ (Figure 49(a) in Appendix G.2). b) The PCA anomaly detection results on benign and adversarial sample embeddings show it cannot detect ~80% of adversarial samples (Appendix G.2 contains detailed results).
> 4. __Error Bar:__ The value of the _best_ error bar is 0, which is why it doesn't have a _top_ or _bottom_. We kept it for consistency but have now removed it (Figure 6).
> 5. __Anomaly Detection as Defense:__ Anomaly detection is an existing defense approach in unsupervised settings. We wanted to show that despite using SOTA deep learning anomaly detection models (SSD), this is not a good enough defense for adversarial attacks against deep clustering. We hope that through our work better defenses are proposed for deep clustering.
> 6. __Queries Table:__ The query complexity table illustrates how costly the proposed attack is. Our goal was to demonstrate that query complexity for the attack is not large, even compared to attacks on non-deep clustering ML tasks. We can reword this to make it clearer in the final paper.
>
> References:
> 1) G Hinton, et al. Distilling the knowledge in a neural network. 2015.
> 2) N Carlini, et al. Towards evaluating the robustness of neural networks. 2017.
> 3) B Yang, et al. Towards k-means-friendly spaces: Simultaneous deep learning and clustering. 2017.
> 4) L Van der Maaten, et al. Visualizing data using t-sne. 2008.
> 5) Y Li, et al. Contrastive clustering. 2021.

---

> > ### Comment · Reviewer_c5rd · 2022-08-08
> > **Response to the authors**
> >
> > Thanks for the additional experiments and further improvements.
> >
> > Your clarifications in Items 2, 3, 4 has well-resolved our concerns. Regarding Items 5 and 6, we have got your explanations. Thanks.
> >
> > In our process of reviewing, one issue that concerns us is how to evaluate your contribution. We agree with what you have claimed on the performance of attack. However, such a successful attack seems not telling anything interesting for development/improvement of deep clustering models. Besides of the success of attacking, we fail to learn anything in-depth from your experimental results.
> >
> > We appreciate your work while it is around the borderline of NeurIPS. So we keep the score of 5.

---

> > > ### Author Response · Authors · 2022-08-09
> > > **Author Reply**
> > >
> > > Thank you for your feedback and for taking the time to go through the revision, we appreciate it.

---

### Author Response · Authors · 2022-08-02
**Summarizing Changes Made During Rebuttal Phase**

We thank the reviewers for their comments and questions. We appreciate all the feedback and suggestions, which further improve our paper.

For easy reference, we have summarized the main changes made to the paper during the rebuttal phase:
- Updated Figure 6 in the main paper to not have a 0 error bar (Reviewer c5rd).
-  Added Appendix G which contains new defense experiments (Appendices G.1 and G.2) using an older deep clustering method, t-SNE, and PCA (Reviewer c5rd and Reviewer hxny).
- Added Appendix E which contains empirical results and comparison with SPSA attack (Reviewer sgLu).
- Added subsection "Attacks/Defenses Against Supervised Learning" to Section 2, Related Works (Reviewer sgLu).
- Added Appendix F which contains more discussion on why supervised attacks/defenses are not applicable to our setting (Reviewer sgLu).
- Corrected the caption for Figure 1 (Reviewer sgLu).
- Reworded footnote on line 35 to avoid ambiguity (Reviewer sgLu).
- Changed subsection "Adversarial Attacks Against Clustering" to "Training-time Adversarial Attacks Against Clustering" to avoid ambiguity (Reviewer sgLu).
- Reworded statement on line 138; this is line 153 in revised submission (Reviewer sgLu).
- Added Appendix H which contains descriptions of 2 real-world practical attack scenarios for our attack setting (Reviewer sgLu and Reviewer 88S9).
- Improved "Limitations" subsection in Section 7 (Reviewer sgLu and Reviewer 88S9).
- Added footnote 2 on line 46 for threat model details (Reviewer 88S9).
- Added lines 84-85 including footnote 4 detailing percentage performance changes (Reviewer 88S9).
- Added footnote 6 on line 145 regarding distinction between score-based and decision-based attack (Reviewer 88S9).

---

### Meta-Review · Area_Chair_WxMN · 2022-08-23

**Recommendation:** Accept
**Confidence:** Certain

**Metareview:**

To investigate the adversarial attacks and robustness for deep clustering models, the authors propose a blackbox attack using Generative Adversarial Networks (GANs) where the adversary does not know which deep clustering model is being used, but can query it for outputs.
Based on several rounds of discussions between the authors and reviewers, the reviewers' concerns have been properly addressed.
Since all reviewers consistently gave positive comments, the AC made a final decision of acceptance.




**Award:**

No

---

### Decision · Program_Chairs · 2022-09-14

Accept